# Exploiting the entire near-infrared spectral range to improve the detection of methane plumes with high-resolution imaging spectrometers

Javier Roger[1], Luis Guanter[1,2], Javier Gorroño[1], and Itziar Irakulis-Loitxate[1,3]

[1]Research Institute of Water and Environmental Engineering (IIAMA), Universitat Politècnica de València (UPV), 46022, Valencia, Spain.
[2]Environmental Defense Fund, Reguliersgracht 79, 1017 LN Amsterdam, The Netherlands.
[3]International Methane Emission Observatory (IMEO), United Nations Environment Programme, Paris, France.

**Correspondence:** Javier Roger (jarojua@upvnet.upv.es)

**Abstract.**

Remote sensing emerges as an important tool for the detection of methane plumes emitted by so-called point sources, which are common in the energy sector (e.g., oil and gas extraction and coal mining activities). In particular, satellite imaging spectroscopy missions covering the shortwave infrared part of the solar spectrum are very effective for this application. These instruments sample the methane absorption features at the spectral regions around 1700 and 2300 nm, which enables the retrieval of per-pixel methane concentration enhancements. Data-driven retrieval methods, in particular those based on the matched filter concept, are widely used to produce maps of methane concentration enhancements from imaging spectroscopy data. Using these maps enables the detection of plumes and the subsequent identification of active sources. However, retrieval artifacts caused by particular surface components may sometimes appear as false plumes or disturbing elements in the methane maps, which complicates the identification of real plumes. In this work, we use a matched filter that exploits a wide spectral window (1000-2500 nm) instead of the usual 2100-2450 nm window with the aim of reducing the occurrence of retrieval artifacts and background noise. This enables a greater ability to discriminate between surface elements and methane. The improvement in plume detection is evaluated through an analysis derived from both simulated data and real data from areas including active point sources, such as the O&G industry from San Joaquin Valley (U.S.) and the coal mines from the Shanxi region (China). We use datasets from the PRISMA and EnMAP satellite imaging spectrometers missions and from the airborne AVIRIS-NG instrument. We obtain that the interference with atmospheric carbon dioxide and water vapor is generally almost negligible, while coemission or overlapping of these trace gases with methane plumes leads to a reduction of the retrieved concentration values. Attenuation will also occur in the case of methane emissions situated above surface structures that are associated with retrieval artifacts. The results show that the new approach is an optimal trade-off between the reduction of background noise and retrieval artifacts. This is illustrated by a comprehensive analysis in a PRISMA dataset with 15 identified plumes, where the output mask from an automatic detection algorithm show an important reduction in the number of clusters not related to $CH_4$ emissions.

# 1  Introduction

Since pre-industrial times, the concentration of methane (CH$_4$) in the atmosphere has increased by more than 150% to a
globally-averaged value of 1920 parts-per-billion (ppb) in early 2023 (Dlugokencky, 2023). CH$_4$ is the second most important
anthropogenic greenhouse gas and has been estimated as responsible for almost a third of the warming of the planet so far
(Masood et al., 2021). In addition, because of its short lifetime and its a relatively fast mitigation potential (Ocko et al., 2021),
reducing atmospheric CH$_4$ concentration is the most efficient way to curb global warming (Ming et al., 2022).

A great portion of the increase in CH$_4$ concentration in the atmosphere is due to the growth of anthropogenic emissions from
sectors such as agriculture, waste management, coal mining and the oil and gas (O&G) industry. The O&G industry produces
approximately 33% of total anthropogenic emissions and has been identified as one of the sectors with the highest potential to
reduce emissions (UNEP, 2021), which is also generally considered cost-effective (Mayfield et al., 2017).

An important fraction of the emissions from fossil fuels comes from CH$_4$ point sources (Duren et al., 2019). In this con-
text, satellites have proven to be instrumental in detecting CH$_4$ plumes originated in this manner. Especially, satellite imaging
spectrometers can leverage the CH$_4$ absorption features in the shortwave infrared (SWIR), where there is a weak and a strong
absorption window around the 1700 nm and 2300 nm spectral wavelengths ($\lambda$), respectively (see Figure 1). Spaceborne mea-
surements of the solar radiation reflected by the Earth's surface can be used to derive CH$_4$ concentration enhancements from
these features.

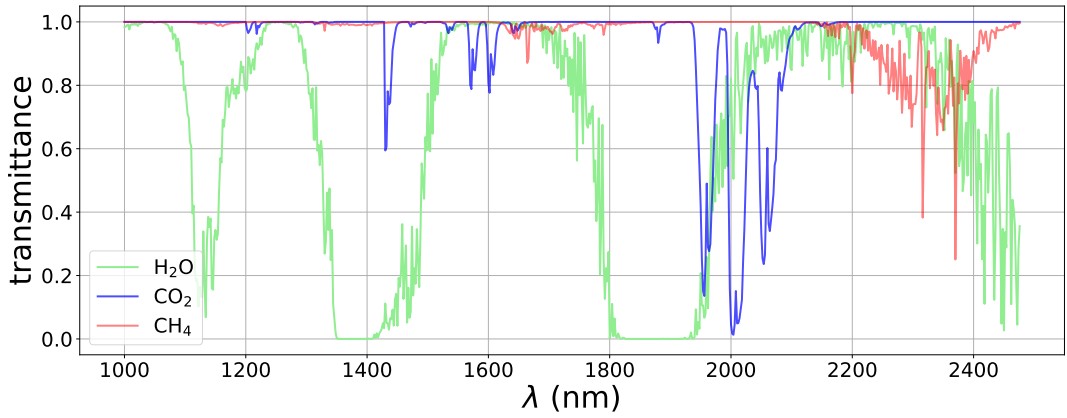

**Figure 1.** MODTRAN derived $transmittance$ spectra of atmospheric H$_2$O (green), CO$_2$ (blue), and CH$_4$ (red) resampled to 2 nm spectral
resolution.

Imaging spectrometers, also known as hyperspectral imagers, have a coarse temporal resolution but offer a spatial and
spectral resolution able to resolve a large range of point sources (Irakulis-Loitxate et al., 2021). Of this kind, we highlight the
Italian PRISMA (Precursore IperSpettrale della Missionse Applicativa) mission (Loizzo et al., 2018) and the German EnMAP
(German Environmental Mapping and Analysis Program) mission (Guanter et al., 2015), both with a 30 m spatial resolution,

a spectral resolution $\sim$10 nm, and a spectral coverage of 400-2500 nm. We also acknowledge the constellation of GHGSat, which have a spatial resolution varying between 25 and 50 m (depending on the satellite). These satellites have built-in Fabry-Pérot spectrometers that operate only in the 1700 nm $CH_4$ absorption window with a spectral resolution of 0.1 nm (Jervis et al., 2021).

In the existing literature, two types of methodologies have been employed for deriving $CH_4$ concentration maps. The first category consists of physically-based methods, which require a comprehensive understanding of radiation, its interactions, and the properties of the medium through which it propagates. On the other hand, data-driven methods extract information statistically from the image, offering advantages such as reduced computational time and partial compensation for radiometric and spectral errors (Thompson et al., 2015; Guanter et al., 2021). Among the possible methods of this class we highlight the matched filter.

The matched filter applied to the $CH_4$ case maximizes the score on the pixels that most strongly match the $CH_4$ absorption spectrum (Manolakis et al., 2007) convolved to the spectral response of the satellite sensor, and the bands selected from the dataset are usually those covering the 2300 nm absorption window (2100-2450 nm). Unfortunately, the matched filter is prone to disturbing enhancements caused by measurement noise and sensitivity to the surface. Specifically, surface elements with similar absorptive features to $CH_4$ may lead to systematic errors (Thorpe et al., 2013; Ayasse et al., 2018; Guanter et al., 2021). These can be abundantly present in heterogeneous scenes and can complicate or mislead the detection of $CH_4$ plumes. We show an example in Figure 2. The identification of real $CH_4$ plumes is complicated because of the large number of these disturbing elements across the scene. Therefore, it is necessary to develop new procedures that get to remove them, and consequently, improve the ability to detect $CH_4$ emissions.

In this work, we present a matched filter-based retrieval that exploits the whole SWIR spectral region with the aim of improving the detection of $CH_4$ plumes emitted from point-sources. We implement end-to-end simulations from trace gases to test if atmospheric carbon dioxide ($CO_2$) and water vapor ($H_2O$) disturb the $CH_4$ retrieval when expanding the spectral range. Moreover, we present $CH_4$ concentration enhancement maps derived from both simulated and real data. These maps are compared with other state-of-the-art retrieval methods, and the plume detection capability is analyzed.

## 2 Materials and methods

### 2.1 Matched filter for $CH_4$ enhancement concentration mapping

If radiation arrives at the detector without $CH_4$ absorption from an emission ($\mathbf{L_{NoAbs}}$) and also with $CH_4$ absorption ($\mathbf{L_{Abs}}$), according to the Beer-Lambert's law, the radiance spectrum $\mathbf{L_{Abs}}$ will be characterized by

$$\mathbf{L_{Abs}} = \mathbf{L_{NoAbs}} \cdot e^{(\Delta \text{XCH}_4 \cdot \mathbf{K_{CH_4}})} \tag{1}$$

where $\Delta\text{XCH}_4$ is the $CH_4$ column concentration enhancement in ppb, and $\mathbf{K_{CH_4}}$ is the $CH_4$ unit absorption spectrum, which characterizes $CH_4$ absorption features. Note that bold font is used to denote vectors. Moreover, $\mathbf{K_{CH_4}}$ is calculated using a

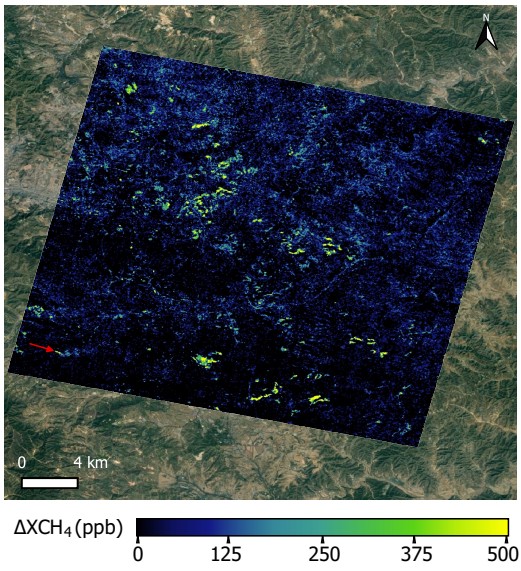

**Figure 2.** Example of a CH$_4$ concentration enhancement ($\Delta$XCH$_4$) map derived from a PRISMA dataset that covers an area of coal mines in the Shanxi region (China). The only detected plume is pointed out with a red arrow and is originated from a venting shaft. The retrieval is derived from a matched filter formulation using the 2100-2450 nm CH$_4$ absorption window. The true color image source is © Google Earth.

LookUp Table adapted to the angular configuration of the scene that relates CH$_4$ transmittance spectra from the MODTRAN (Spectral Sciencies, Inc., 2016) radiative transfer code to CH$_4$ column concentration values. We do not consider atmospheric scattering in the calculation of transmittance for the SWIR spectral range, assuming a pristine atmosphere. Nevertheless, in certain cases such as in the presence of dust particles or in dark surfaces with low solar zenith angles, the influence of scattering can be significant (Thorpe et al., 2014). Note that $\mathbf{K_{CH_4}}$ in satellite-based missions is calculated considering the integration of CH$_4$ over an 8 km high column such as in Thompson et al. (2016), while in airborne missions is calculated over the specific flight height. Then, the exponential of the Eq. 1 can be expanded as a Taylor series of infinite terms, and it can be simplified to the second term if we assume a sufficiently small argument of the exponential. This implies that the approximation will generate more accurate results for lower $\Delta$XCH$_4$. Then, assuming this simplification, radiance can be approximated to a linear function of $\Delta$XCH$_4$, which will be used for the matched filter method.

The matched filter models a scene radiance data cube as a multivariate Gaussian, where each spectral band is considered to follow a Gaussian distribution. The spectral mean vector ($\boldsymbol{\mu}$) and the covariance matrix ($\Sigma$) are retrieved from the data cube and characterize the whole image, while assuming enough homogeneity and CH$_4$ emission sparsity through the scene. Then, the radiance spectrum of each pixel ($\mathbf{L}$) can be assessed following two different hypotheses. The null hypothesis ($H_0$), where

radiance is simply assessed as background radiance, and the alternative hypothesis ($H_1$), where it is assessed as the background radiance plus a term that represents CH$_4$ absorption (Thompson et al., 2016). These hypothesis are represented as follows

$$H_0 : \mathbf{L} \sim \mathcal{N}(\boldsymbol{\mu},\, \Sigma) \tag{2}$$

$$H_1 : \mathbf{L} \sim \mathcal{N}(\boldsymbol{\mu} + \Delta\text{XCH}_4 \cdot \mathbf{t},\, \Sigma) \;=\; \mathcal{N}(\boldsymbol{\mu}(1 + \Delta\text{XCH}_4 \cdot \mathbf{K_{CH_4}}),\, \Sigma) \tag{3}$$

where $\mathcal{N}$ represents a multivariate Gaussian distribution with the mean vector and covariance matrix located in its first and second arguments, respectively. The CH$_4$ absorption term from $H_1$ is given by the linear term $\Delta\text{XCH}_4 \cdot \mathbf{t}$, that comes from linearizing the exponential function from Eq. 1. $\mathbf{t}$ is the target signature that spectrally characterizes the absorption of CH$_4$ per unit of concentration and is obtained by an element-wise multiplication between the $\boldsymbol{\mu}$ and the $\mathbf{K_{CH_4}}$ arrays, where $\mathbf{K_{CH_4}}$ is first convolved to the instrument's spectral response.

In order to obtain the $\Delta\text{XCH}_4$ values, the probability of $H_1$ occurring is maximized following the maximum likelihood estimation (Eismann, 2012). As a result, we obtain this expression

$$\Delta\text{XCH}_4 = \frac{(\mathbf{L} - \boldsymbol{\mu})^T \Sigma^{-1} \mathbf{t}}{\mathbf{t}^T \Sigma^{-1} \mathbf{t}} \tag{4}$$

Datasets used in this study come from push-broom imaging spectrometer missions (AVIRIS-NG, EnMAP, and PRISMA) that scan the swath of the scene with a 2-D detector array. Differences in central wavelength and spectral resolution can exist among detectors from the same array because of optical aberrations (Guanter et al., 2009), which compromises the uniformity between the data cube columns in the across-track direction. Therefore, the matched filter is applied for each along-track column separately.

Along the SWIR spectral window there are two CH$_4$ absorption windows, namely a weaker one around 1700 nm and a stronger one around 2300 nm. The 2300 nm window (roughly 2100-2450 nm) is typically chosen for CH$_4$ mapping (Thompson et al., 2015; Foote et al., 2020; Guanter et al., 2021; Irakulis-Loitxate et al., 2021) because the more intense absorption and spectral sampling allows to better characterize CH$_4$ in comparison to the other window. Hereinafter we will refer this spectral range selection as 2300-MF. There are usually elements from the scene such as roads, solar panels, and buildings that present absorption features similar to CH$_4$ in the 2300 nm window. Therefore, we will find retrieval artifacts, i.e., pixels with positive $\Delta\text{XCH}_4$ values related to these structures. Retrieval artifacts usually present higher $\Delta\text{XCH}_4$ values than background noise and can disturb CH$_4$ plume detection. In fact, they complicate the identification of the real plumes and can even lead to false positives. Now, we examine the functionality of the matched filter to minimize the presence of retrieval artifacts.

In Figure 3, we show a scatter plot between two spectral bands with CH$_4$ absorption (2371 nm) and with no CH$_4$ absorption (2102 nm) from a PRISMA dataset covering a homogeneous arid area in Sudan. The $\Delta\text{XCH}_4$ values resulting from applying the matched filter using only these two bands are illustrated following the color gradient on the right side of the figure. Note that

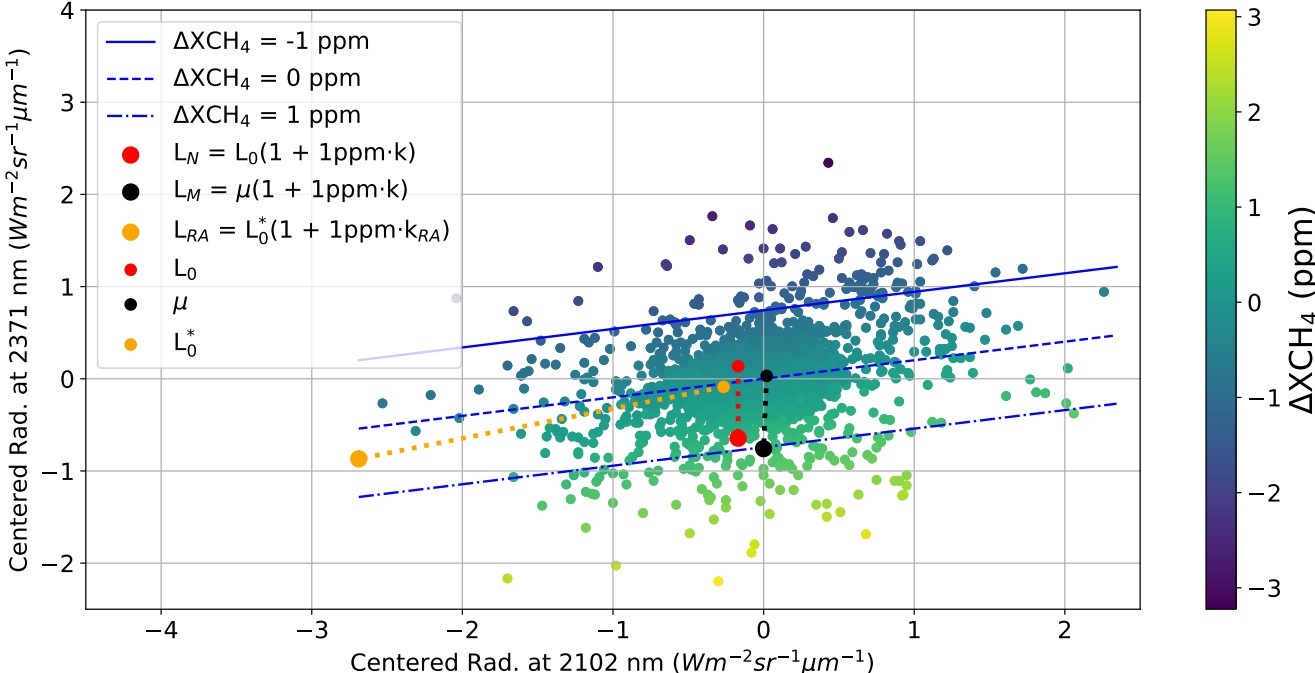

**Figure 3.** Scatter plot between the centered radiance data points from the spectral band with no $CH_4$ absorption (2102 nm) and with $CH_4$ absorption (2371 nm) from a PRISMA dataset. The corresponding $\Delta XCH_4$ values result from the matched filter applied to the two spectral bands and are plotted following the colorbar at the right. The blue lines indicate the domain in where the data points have $\Delta XCH_4$ values of -1 (top), 0 (center) and 1 (bottom) parts-per-million. Small dots from the legend are the original data points before the effective absorption of 1 ppm (big dots) for the natural (red), modeled (black) and the retrieval artifact (orange) cases.

the radiance data points are plotted relative to the mean values, highlighting the impact of shifts from these central values in the retrieved data. A linear fit based on these shifts allows us to determine the lines equivalent to the $\Delta XCH_4$ values of -1, 0, and 1 parts-per-million (ppm). Moreover, three pixels from the original dataset were selected and transformed to represent different situations: one with a 1 ppm natural absorption, another with a 1 ppm absorption according to the matched filter model, and a third to illustrate the presence of a retrieval artifact. The absorption based on the model aligns with the alternative hypothesis $H_1$ from Eq. 3, indicating a deviation from the mean radiance spectrum in the target spectrum direction. Instead, the natural absorption, which reflects a realistic $CH_4$ absorption, is implemented by deviating the radiance spectrum of a randomly chosen pixel in the dataset also in the target spectrum direction. On the other hand, the retrieval artifact was generated similarly to the natural absorption case but with an additional deviation in the non-absorption band, where the target spectrum has a null value. The radiance spectra related to these three cases are expressed as follows:

$$\mathbf{L_N} = \mathbf{L_0}(1 + 1ppm \cdot \mathbf{K_{CH_4}}) \tag{5}$$

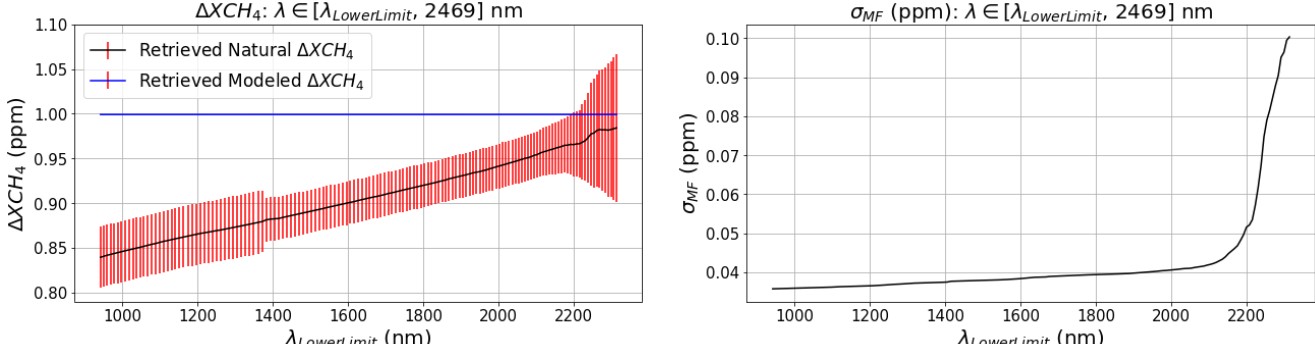

**Figure 4.** In the left panel, the mean $\Delta XCH_4$ values related to the retrieved natural (black) and modeled (blue) artificial absorptions applied to 1000 different pixels from a PRISMA dataset and their associated error bars (red). In the right panel, a standard deviation $\sigma_{MF}$ resulting from the Gaussian distribution that is followed by the dataset retrieved values. The matched filter spectral range of application ranges from a variable wavelength value ($\lambda_{LowerLimit}$) to 2469 nm.

$$\mathbf{L_M} = \boldsymbol{\mu}(1 + 1ppm \cdot \mathbf{K_{CH_4}})$$ (6)

$$\mathbf{L_{RA}} = \mathbf{L_0^*}(1 + 1ppm \cdot \mathbf{K_{RA}})$$ (7)

where $\mathbf{L_N}$, $\mathbf{L_M}$, and $\mathbf{L_{RA}}$ are the radiance spectra vectors referring to a 1 ppm natural absorption, a 1 ppm modeled absorption, and a retrieval artifact, respectively. $\mathbf{L_0}$ and $\mathbf{L_0^*}$ are the selected radiance spectra vectors from the original dataset pixels, and $\mathbf{K_{RA}}$ is a made up equivalent unit absorption spectrum for the retrieval artifact. The difference between $\mathbf{K_{RA}}$ and $\mathbf{K_{CH_4}}$ is that $\mathbf{K_{RA}}$ presents a non-zero value in the band with no $CH_4$ absorption. While the modeled absorption in $\mathbf{L_M}$ results in 1 ppm, the natural absorption in $\mathbf{L_N}$ suffers from underestimation due to the deviation of the original pixel radiance $\mathbf{L_0}$ from the mean. Enhancements values associated to natural absorption scatter around their true value since the measurements are subject to noise, thus an overestimation would have been equally likely. Note that in Figure 3, the retrieval is expected to yield a wide range of $\Delta XCH_4$ values due to the limited number of bands involved. On the other hand, while the retrieval artifact $\mathbf{L_{RA}}$ exhibits a spectral feature similar to $CH_4$ in the absorption band, it also displays a characteristic deviation in the non-absorption band that enables better discrimination from $CH_4$. Thus, by expanding the spectral window in the matched filter application, a more comprehensive discrimination between retrieval artifacts and $CH_4$ may lead to their attenuation or complete removal.

Additional variations may arise by expanding the window, as depicted in Figure 4. For 1000 different pixels from the already employed PRISMA dataset, absorptions of 1 ppm were applied using modeled and natural absorptions with Eq.6 and Eq.5, respectively. In the right panel, we show the standard deviation resulting from the entire matched filter retrieval values when

using a spectral window comprising a variable lower limit ($\lambda_{LowerLimit}$) and 2469 nm. We select this upper limit because it allows to cover the entire 2300 nm $CH_4$ absorption window. Incorporating more bands implies a more demanding spectrum-to-match, and the covariance matrix will better suppress the background, which will result in a lower standard deviation in the retrieval. In the left panel, we display the mean $\Delta XCH_4$ values retrieved from the pixels with the modeled (blue) and natural (black) absorptions, along with their related standard deviation values (red) as a measure of uncertainty. While modeled absorption consistently retrieves the expected 1 ppm value with negligible uncertainty, natural absorption yields progressively lower values when expanding the window. Similar to the background pixels, the accumulated deviations from the natural absorption compared to the modeled absorption will result in a reduction of the retrieved enhancement. As we can see, incorporating more bands makes the matched filter more sensitive to spectrum changes, which leads to a background noise reduction and an underestimation of the $\Delta XCH_4$ values from plume pixels. This increased sensitivity could also capture undesired variations from the modeled spectrum due to factors unrelated to methane, such as atypical albedo values or instrument noise, potentially resulting in clutter noise. New retrieval artifacts may appear when including the 1700 nm absorption window, but overall, the increased spectral interval mitigates those from both the 1700 nm and 2300 nm windows efficiently, as is shown in section 3.

We have not considered yet the interference with other trace gases such as the $H_2O$ and $CO_2$ when expanding the window (see Figure 1). Absorption features from these gases appear along the SWIR spectral region and might cause further biases on the $\Delta XCH_4$ values from plume pixels. The left panel of Figure 4 shows a sudden increase of the standard deviation of the retrieved $\Delta XCH_4$ values related to the natural absorption when including several spectral bands around 1400 nm. This occurs because an important fraction of radiance pixels in these bands contains absolute zero values due to strong $H_2O$ absorption, which violates the assumption of Gaussian modeling in the matched filter. If we remove these bands, the standard deviation increase does not occur. Consequently, we exclude the bands around 1400 nm along with those associated with the strong $H_2O$ absorption around 1900 nm, as they may also compromise the retrieval due to their very low values. Moreover, the atmospheric concentrations of $H_2O$ and $CO_2$ are approximately homogeneous across the area captured in one dataset. However, variations in atmospheric concentration of $H_2O$ are given when there are pronounced height variations of the terrain across the scene (Lou et al., 2021). Unless the dataset area meets this condition, we can assume homogeneity. As a result, variations from the mean array cannot be attributed to these gases, which implies that the matched filter should not be affected by them. Nevertheless, the coemission or overlapping of plumes from $H_2O$ or $CO_2$ with $CH_4$ plumes has not been studied yet. Simulations will be employed to understand how they impact the retrieval. This could help to understand methane concentration maps in situations such as inefficient flaring (Irakulis-Loitxate et al., 2021), where there is coemission of $CH_4$ and $CO_2$ plumes.

## 2.2 Combo-MF

In this work, we propose to expand the matched filter spectral range of application to the whole SWIR (1000 – 2500 nm) in order to remove retrieval artifacts and therefore to improve $CH_4$ plume detection. Hereinafter we will call this spectral range selection as SWIR-MF. We select this relatively wide spectral range to leverage the surface-related spectral features, aiming to maximize the mitigation of retrieval artifacts. This selection would be unfeasible using physically-based methods such as the IMAP-DOAS (Frankenberg et al., 2005) because of the growing complexity of separating atmospheric absorptions from

surface spectral features (Thorpe et al., 2014). Also note that the increased number of spectral bands does not substantially impact the computational processing time required when applying the matched filter.

As we observed in Figure 4, SWIR-MF produces overall smaller $\Delta XCH_4$ values than 2300-MF. This is especially true for those retrieval artifacts that are displayed in the 2300-MF retrieval. However, expanding the window could also derive in the appearance of clutter due to the higher matched filter sensitivity and the inclusion of new absorbing bands that could lead to the emergence of new retrieval artifacts. Therefore, SWIR-MF values greater than the 2300-MF values will probably come from disturbing factors. As a solution, we change these values to the ones of the original 2300-MF in order to penalize the increased enhancement. On the other hand, to solve the underestimation from plume pixels and keep the variances of both retrievals comparable, we multiply the remaining retrieval values by a factor $f$ defined as

$$f = \frac{\sigma_{(2300)}}{\sigma_{(SWIR)}} \tag{8}$$

where $\sigma_{(2300)}$ and $\sigma_{(SWIR)}$ are the standard deviations from the Gaussian distributions that are followed by the $\Delta XCH_4$ retrieved values from the dataset resulting from applying the 2300-MF and SWIR-MF retrievals, respectively. Generally, retrievals follow a normal distribution with an averaged value of $\sim 0$, so we can scale the distribution of not-transformed SWIR-MF values to the one of the 2300-MF values by simply multiplying by $f$. Then, the transformed values from the SWIR-MF retrieval will follow approximately the 2300-MF retrieval normal distribution. In this manner, plume pixel values will be approximately scaled to the enhancement levels from the 2300-MF retrieval in order to solve the underestimation. We will call this procedure Combo-MF, which can be expressed as

$$\Delta XCH_{4(Combo)} = \begin{cases} f \cdot \Delta XCH_{4(SWIR)} & \text{if } \Delta XCH_{4(SWIR)} < \Delta XCH_{4(2300)} \\ \Delta XCH_{4(2300)} & \text{if } \Delta XCH_{4(SWIR)} \geq \Delta XCH_{4(2300)} \end{cases} \tag{9}$$

where $\Delta XCH_{4(2300)}$, $\Delta XCH_{4(SWIR)}$, and $\Delta XCH_{4(Combo)}$ are the $CH_4$ column enhancement values from the 2300-MF, SWIR-MF, and Combo-MF retrievals, respectively. As a result, limitations from the SWIR-MF are mitigated: the emergence of new clutter noise is removed, and the SWIR-MF values are transformed into typical 2300-MF value levels. Nevertheless, in pixels where surface structures related to a retrieval artifact are positioned beneath a $CH_4$ emission, there might be a more pronounced attenuation that could compromise the detection. Figure 5 shows retrieval histograms from an EnMAP dataset from an O&G field in Turkmenistan when applying 2300-MF, SWIR-MF, and Combo-MF. We observe that SWIR-MF has a lower standard deviation because of a more stringent spectrum that reduces background noise. The Combo-MF takes a standard deviation close to the one from the 2300-MF because of the scaled values, but its mean is shifted to negative values because a great fraction of the 2300-MF values from Combo-MF are negative. Therefore, there is a lower amount of pixels with positive enhancement, which implies a better contrast between plume pixels and their background.

Combo-MF is not the unique alternative created to improve $CH_4$ emission detection. Instead, there exist other matched filter derived methods such as the *Matched filter with Albedo correction and reweighted L1 sparsity code* (MAG1C) that has

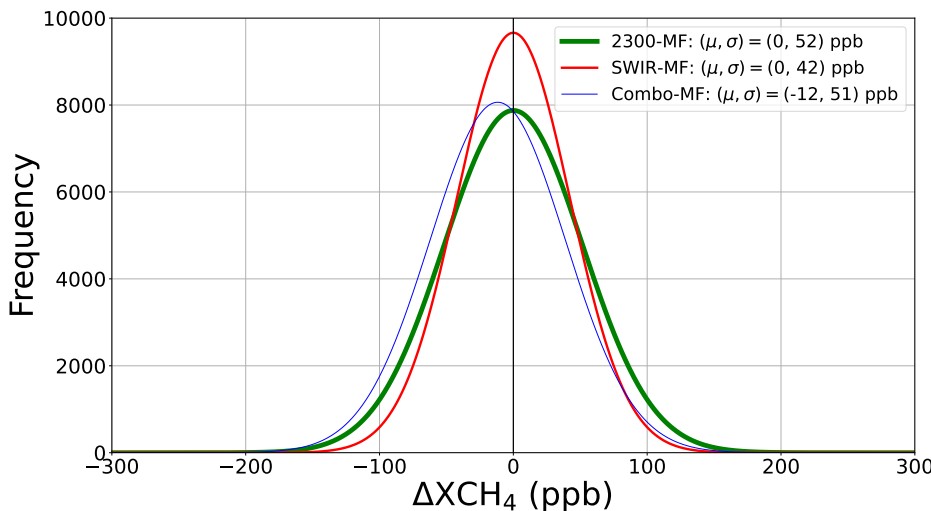

**Figure 5.** Histrogram from 2300-MF (green), SWIR-MF (red), and Combo-MF (blue) retrieval histograms from an EnMAP dataset showing a O&G field located in Turkmenistan with $CH_4$ emissions. $\mu$ and $\sigma$ are the mean and standard deviation values from the different distributions, respectively.

been used in some studies (Foote et al., 2021; Ayasse et al., 2022; Knapp et al., 2023). It applies an albedo correction across the radiance data cube in order to account for the homogeneity assumption from the matched filter. In addition, MAG1C leverages the $CH_4$ sparsity assumption and also applies an iterative regularization that aims to reduce background noise. MAG1C retrievals values can be divided in two groups: 1) zero-values that were obtained because of the sparsity assumption and 2) a set of values that follow a log-normal distribution and mostly refer to retrieval artifacts and $CH_4$ emissions. If we transform the retrieval in order to follow a normal distribution, the resulting standard deviation will not be related to the random background noise, as happens with Combo-MF, and therefore we will not be able to compensate a potential underestimation when extending the spectral window. Because of this reason, we will not apply an equivalent Combo-MF methodology to MAG1C. Thus, we will compare MAG1C retrievals using the 2300 nm absorption window (2300-MAG1C) and the whole SWIR spectral region (SWIR-MAG1C) to the Combo-MF retrievals in order to assess if the latter improves plume detection.

### 2.3 Simulated trace gas enhancements

$\Delta XCH_4$ maps depicting simulated plumes with different shapes, related wind speed values, and concentrations have been generated using large-eddy simulations with the Weather and Research Forecasting Model (WRF-LES) (Varon et al., 2018; Cusworth et al., 2019). Using the deduced LookUp Table used to obtain $\mathbf{K_{CH_4}}$, the $\Delta XCH_4$ values associated with the synthetic plumes are transformed into their equivalent transmittance spectra. Then, transmittance is convolved to the instrument's spectral response and multiplied by the original radiance dataset, accounting for minor radiometric offsets (Guanter et al.,

2021). These plumes are implemented in three different datasets from the PRISMA mission to assess the $CH_4$ emission detection and quantification capabilities from the different procedures.

Simulations are also used to assess the impact of $H_2O$ and $CO_2$ on the retrievals. Simulated enhancements of typical background concentrations for $CO_2$ (2500–10000 ppb) (C3S, 2018) and $H_2O$ (1–2.5 g/cm$^2$) (Mieruch et al., 2014) are uniformly applied to the whole extent of a PRISMA dataset with a constant value. This allows us to examine how variations in the atmospheric concentrations of these trace gases affect $CH_4$ retrievals using 2300-MF and Combo-MF. Additionally, we introduce $H_2O$ and $CO_2$ enhancements exclusively on pixels where we implemented a simulated $CH_4$ plume to assess the impact of these trace gases on the plume $\Delta XCH_4$ values in the case of coemission or overlapping plumes. A constant enhancement of 10 ppm has been applied for $CO_2$, and 100 ppm for $H_2O$, considering the order of magnitude observed in some $CO_2$ and $H_2O$ plumes reported by Cusworth et al. (2023) and Thorpe et al. (2017), respectively.

## 2.4 Detection and quantification of point-source emissions

Generally, the detection of $CH_4$ plumes is performed through a supervised methodology, which involves a direct visual inspection of the retrieval. First, we search for clusters with plume-like shapes and with $\Delta XCH_4$ values greater than background noise values. Then, we validate the association with a potential $CH_4$ source using high-resolution images from Google Earth data or the spectral radiance bands within the dataset. Finally, we confirm the cluster as a true plume by validating its alignment with the wind direction, which can be retrieved from the GEOS-FP database (Molod et al., 2012). For instance, the emission in Figure 2 was confirmed as the only plume in the image according to these steps. However, in order to establish a systematic procedure to study the detection capabilities of both the 2300-MF and Combo-MF, we will use an automatic detection algorithm instead of the supervised initial selection of clusters associated with potential plumes. The algorithm begins by applying a 3 x 3 median filter to the retrieval to mitigate the impact of random noise in the classification process (Varon et al., 2018). Subsequently, a mask is generated, retaining only those values from the filtered retrieval that exceed 1 standard deviation from the Gaussian distribution followed by the originally retrieved values. It is assumed that this threshold is reasonable for effectively distinguishing between background and plume pixels (Guanter et al., 2021). Next, a filter based on morphological parameters is applied to the resulting mask, isolating clusters with plume-like shapes. Finally, an analysis following the remaining supervised detection steps is applied to the persisting clusters, keeping only those related to $CH_4$ plumes.

On the other hand, the parameter typically used to quantify the intensity of $CH_4$ emissions released from a point-source is the flux rate ($Q$ in $\mathrm{kg/h}$). As in Varon et al. (2018), we can express this magnitude as

$$Q = \frac{U_{\mathrm{eff}} \cdot \mathrm{IME}}{L} \tag{10}$$

where $U_{\mathrm{eff}}$ (m/s) is the effective wind speed, a parameter that linearly depends on the wind speed at 10 m above the surface ($U_{10}$) for PRISMA and EnMAP data (Guanter et al., 2021; Roger et al., 2023), $L$ (m) is obtained as the square root of the plume mask area, and IME (kg) is the mass related to the $CH_4$ enhancement contained in the masked plume (Frankenberg et al., 2016). IME and $L$ depend on the plume mask, which makes masking an important factor that impacts on the accuracy of

**Table 1.** Information about the datasets used in this study. They are listed in order of appearance in this work. Dates are in YYYY-MM-DD format, and latitude and longitude coordinates (*Lat/Lon*) are in decimal degrees.

| Mission | Location | Sector | Date | Lat/Lon |
|---|---|---|---|---|
| PRISMA | Shanxi (China) | Coal mining | 2020-04-28 | 37.831/113.701 |
| PRISMA | Northern State (Sudan) | Not related to CH$_4$ | 2020-04-01 | 21.900/28.000 |
| EnMAP | Ekizak (Turkmenistan) | O&G | 2022-10-02 | 38.685/54.243 |
| PRISMA | Silesia (Poland) | Coal mining | 2022-03-02 | 50.110/17.895 |
| PRISMA | In Amenas (Algeria) | O&G | 2021-01-15 | 28.286/9.638 |
| PRISMA | Permian Basin (U.S.) | O&G | 2020-06-30 | 31.437/-103.480 |
| AVIRIS NG | San Joaquin Valley (U.S.) | O&G | 2017-09-06 | 35.279/-119.476 |
| EnMAP | Delhi (India) | Landfills | 2023-04-24 | 28.620/77.200 |
| PRISMA | Shanxi (China) | Coal mining | 2021-02-06 | 36.241/112.909 |

quantification. Masking is performed with the automatic algorithm used for detection. The resulting masks from both 2300-MF and Combo-MF retrievals may differ due to variations in background noise and the presence of retrieval artifacts. Moreover, the $\Delta$XCH$_4$ values of Combo-MF plume pixels cannot be trusted for quantification because the factor $f$ used is only valid for scaling to typical 2300-MF values, and it does not ensure the accuracy of the enhancement values. To assess whether the masking from Combo-MF improves quantification compared to 2300-MF, we will implement a quantification strategy using Combo-MF masking and 2300-MF values, which we call Mix-MF. This comparison is performed on multiple simulated plumes with known $Q$ and $U_{eff}$ values that have been applied to real datasets. A preliminary mask, delineating the plume shape coming directly from the simulation data, is implemented to eliminate the influence of retrieval artifacts, which allows the isolation of background noise as the only disturbing contribution to quantification. This is also done to automatize the quantification process due to the large number of simulated plumes used for this analysis.

## 2.5 Imaging spectroscopy data

In this work, we have used Top-of-Atmosphere radiance datasets from the PRISMA, EnMAP, and AVIRIS-NG missions. A PRISMA dataset from a coal mining region in Shanxi (China) is selected to exemplify the difficulties in emission detection in CH$_4$ retrievals when using 2300-MF (Figure 2). In this case, only one plume could be detected among a high number of retrieval artifacts. Another PRISMA dataset from a Sudan area is used to illustrate how the matched filter scores background,

retrieval artifacts, and plume pixels by introducing artificial absorptions and retrieval artifacts (Figure 3). It is also used to assess its performance when expanding the spectral window of application (Figure 4). This dataset shows a very low surface variability and no emissions, simplifying the analyses. Additionally, for implementing simulated enhancements of different trace gases, we selected PRISMA datasets from an arid O&G field in Argelia, an O&G field in the Permian Basin (U.S.), a coal mine site in Poland, and an EnMAP dataset from an O&G field in Turkmenistan. The latter is also used to show the distribution of $\Delta$XCH$_4$ values when using 2300-MF, SWIR-MF, and Combo-MF retrievals (Figure 5). Note that, although some of the datasets with simulated enhancements present real CH$_4$ emissions, the simulated plumes were intentionally implemented to avoid interference with the actual ones. Moreover, an AVIRIS-NG dataset from an O&G field in San Joaquin Valley in California (U.S.) showing two plumes is used in order to show the performance of 2300-MF, Combo-MF, 2300-MAG1C and SWIR-MAG1C procedures. In addition, we show retrievals from an EnMAP dataset capturing a Delhi (India) area where there is CH$_4$ release originated in the Gazhipur landfill. Finally, we show the detection capabilities of 2300-MF and Combo-MF in another PRISMA dataset from the Shanxi area, where 15 plumes were identified. More information about these datasets can be found in Table 1.

## 3 Results

### 3.1 Real data with simulated trace gas enhancements

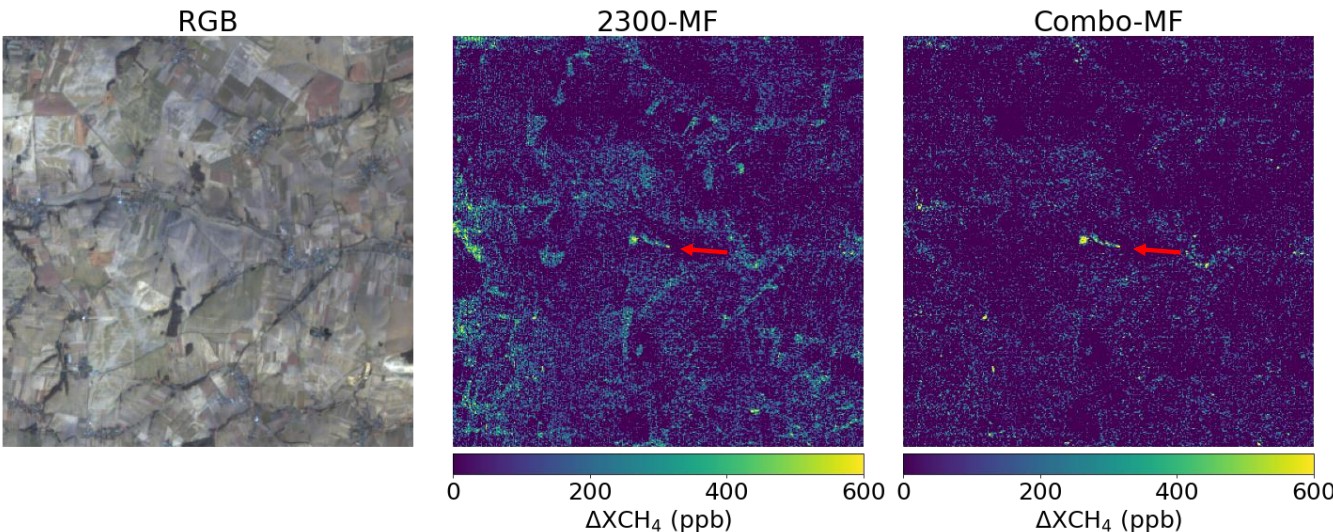

**Figure 6.** True color image (left), 2300-MF retrieval (center), and Combo-MF retrieval (right) from a PRISMA dataset from a coal mine site in Poland with an implemented plume with $Q = 2000\ \mathrm{kg/h}$. The plume is pointed out with an arrow.

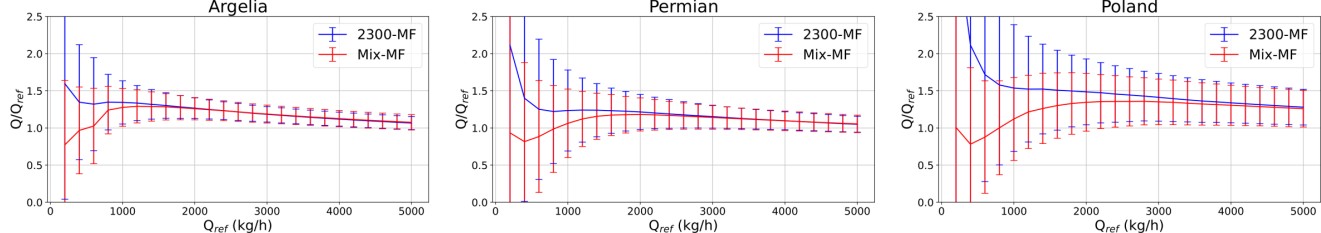

**Figure 7.** Simulated flux rate $Q_{\text{ref}}$ versus the ratio between the deduced flux rate $Q$ and $Q_{\text{ref}}$ from 100 different plumes implemented in PRISMA datasets from sites in Argelia (left), the Permian Basin (center), and Poland (right) using the Mix-MF (red) strategy and the 2300-MF retrievals (blue). Plotted points and error bars of each flux rate value correspond to the mean and 1 standard deviation values from the 100 estimates, respectively.

We have implemented simulated $CH_4$ plumes in three PRISMA datasets from three different sites. These areas were selected to study a diverse range of scenarios regarding surface heterogeneity and brightness, which are important factors in the matched filter retrieval. Homogeneous and bright surfaces will lead to a better performance than more heterogeneous and darker surfaces. The areas selected were an homogeneous and bright arid O&G site from Algeria, an heterogeneous and bright O&G site from the Permian Basin (U.S.), and an heterogeneous and relatively dark coal mining site in Poland.

In Figure 6 we show the RGB image and the 2300-MF and Combo-MF retrievals from the Poland site. We observe a remarkable reduction in the positive background noise values and the attenuation of retrieval artifacts related to the scene topography or to different land covers. As a result, there is a greater contrast between the plume and its surroundings and therefore it is easier to detect the emission by visual inspection. On the other hand, Figure 7 shows the Mix-MF quantification and the one based entirely on the 2300-MF retrieval. The study covers a flux rate interval ranging from 200 kg/h to 5000 kg/h for 5 different plumes with different shape and $\Delta XCH_4$ distribution. Each one of these plumes was implemented in 20 different locations within each dataset and for each flux rate value, i.e., quantification was assessed with 100 plumes for each flux rate value. Thus, we examine the quantification across a diverse range of plumes and backgrounds within the same dataset. Error bars display 1 standard deviation from the quantification distribution related to each implemented flux rate value ($Q_{ref}$). We observe that Argelia and Permian Basin sites present lower uncertainty than the Poland site because the lower brightness from the latter is translated in noisier retrieval. For $Q < 1000$ kg/h, the low $\Delta XCH_4$ plume values, closer to noise level, complicate the masking process. In the 2300-MF quantification case we observe an overestimation, while for the Mix-MF quantification we generally obtain more accurate values. The number of pixels with $\Delta XCH_4$ values greater than 1 standard deviation from the distribution followed by the retrieved values are reduced in Mix-MF due to the clutter removal of Combo-MF, resulting in a more restrictive mask that is less affected by background noise. However, for both procedures there is an important uncertainty at these flux rate levels that limits quantification. For greater $Q$, there is a progressive uncertainty and overestimation reduction for both procedures that approximates quantification to true values. At these levels, estimations from

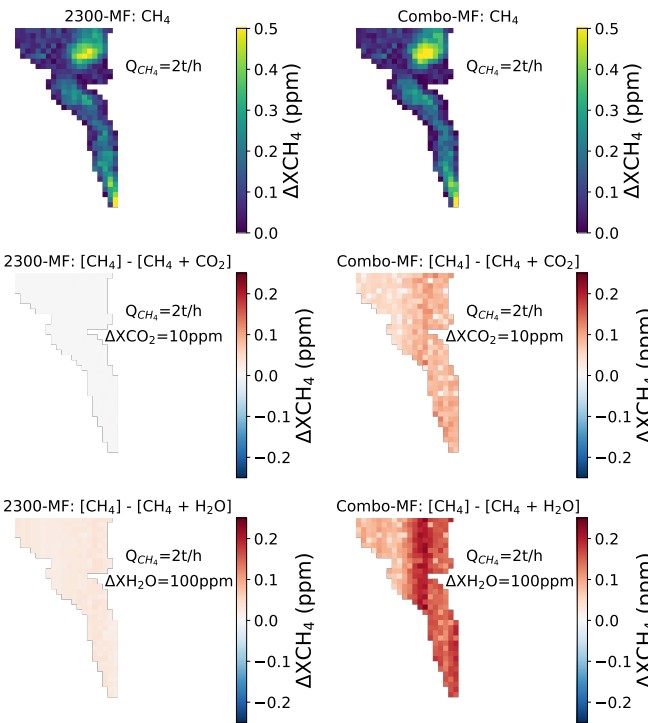

**Figure 8.** Retrieved simulated plume of $Q_{CH_4}$ = 2000 kg/h implemented on an EnMAP dataset from a O&G field in Turkmenistan using 2300-MF (left column) and Combo-MF (right column) retrievals (top row), and the difference resulting from applying only to the plume pixels a constant $\Delta XCO_2$ value of 10 ppm (centered row) and a constant $\Delta XH_2O$ value of 100 ppm (bottom row).

both quantification strategies align because the greater enhancement from plume pixels lead to a practically identical masking for both of them.

In addition, typical background enhancements of $CO_2$ and $H_2O$ were implemented homogeneously across these three datasets in order to assess their impact in 2300-MF and Combo-MF retrievals. As a result, we observe practically null dif-
315    ferences between the enhanced and not enhanced datasets using both procedures. Therefore, this confirms our hypothesis that the atmospheric concentrations of these trace gases do not disturb the retrievals since the changes in radiance are already integrated into the mean value of the scene. On the other hand, we have also studied the influence of $CO_2$ and $H_2O$ enhancements only overlapped with a simulated $CH_4$ plume of $Q$ = 2000 kg/h implemented in an EnMAP dataset covering an O&G area in Turkmenistan. In Figure 8, the top row displays the retrieved $CH_4$ plume obtained using 2300-MF (left column) and Combo-
320    MF (right column). The centered row shows the difference between the original plume and the same plume with an overlapped constant $CO_2$ enhancement ($\Delta XCO_2$) of 10 ppm, and the bottom row presents the difference with an overlapped constant $H_2O$ enhancement ($\Delta XH_2O$) of 100 ppm. In the 2300-MF, we observe that there is a negligible difference when overlapping $CO_2$ enhancements because of the little interference in the 2300 nm window. However, this interference does exist for the $H_2O$ case,

which results in lower retrieved $\Delta$XCH$_4$ values. On the other hand, there is an increased underestimation for both trace gases when expanding the window (Combo-MF) because more absorption features that deviate from the CH$_4$ absorption spectrum are introduced. Nevertheless, if the bands where other trace gases absorb are discarded when expanding the window, there will be no attenuation in the retrieved values due to the absence of interference with the CH$_4$ absorption spectrum, as observed in the CO$_2$ case in the 2300-MF retrieval. Then, when this interference does exist, the degree of underestimation will depend of the overlapped trace gas enhancements levels. Although these enhancements have been applied homogeneously to the plume area, there is an heterogeneous underestimation across the columns that is more pronounced in the H$_2$O case. This occurs because the $\Delta$XCH$_4$ values from the simulated plume and the enhancements from the other traces gases can alter the statistics of the column and therefore resulting in a different underestimation. In addition, the noisy nature from the differences comes mostly from the retrieval noise.

## 3.2 Real CH$_4$ plume cases

### 3.2.1 Comparison to MAG1C retrievals with AVIRIS-NG data

In Figure 9, we compare $\Delta$XCH$_4$ retrievals using 2300-MF, Combo-MF, 2300-MAG1C, and SWIR-MAG1C from an AVIRIS-NG dataset capturing an O&G field site in San Joaquin Valley (U.S.), where two plumes can be observed. 2300-MAG1C retrieval presents higher $\Delta$XCH$_4$ plume values as shown in Guanter et al. (2021) and also exhibits a greater number of retrieval artifacts than 2300-MF. These artifacts are further attenuated or totally removed in the SWIR-MAG1C retrieval. However, SWIR-MAG1C enhances some retrieval artifacts that are not shown when using Combo-MF. The latter presents a lower noise compared to 2300-MF, although it is slightly higher than in MAG1C, which leverages the sparsity assumption for its suppression. Therefore, Combo-MF can be considered as an effective trade-off between reducing background noise and retrieval artifacts. Note that, instead of considering an integration of CH$_4$ over a 8 km high column as with satellite-based data, we use the scene average sensor altitude (2.48 km) from the airborne AVIRIS-NG instrument at acquisition time.

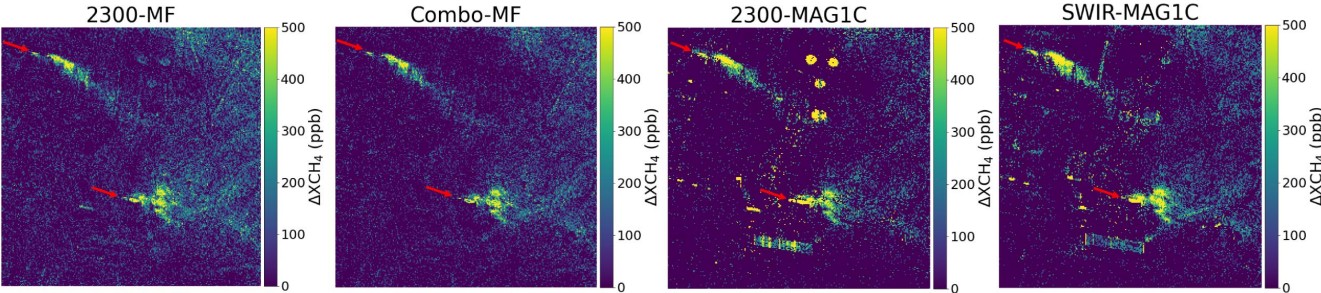

**Figure 9.** Starting from the left, 2300-MF, Combo-MF, 2300-MAG1C, and SWIR-MAG1C retrievals from an AVIRIS-NG dataset showing an O&G field in San Joaquin Valley, California (U.S.). Plumes are pointed out with an arrow.

### 3.2.2 Retrieval performance in a landfill area using EnMAP data

In Figure 10, the 2300-MF (top-right) and Combo-MF (bottom-right) retrievals from an EnMAP dataset capturing a Delhi area (India) are compared. Both retrievals show an emission (framed in red) coming from the Gazhipur landfill. While 2300-MF shows a great amount of retrieval artifacts related to the urban area and a higher background noise, Combo-MF further attenuates or removes these retrieval artifacts and reduces the positives values from background noise. The difference between both retrievals (bottom-left) shows us that there is also attenuation in those pixels where we detect $CH_4$ emission. In fact, we can identify this attenuation in the landfill area in the Combo-MF retrieval. This could be caused by the existence of retrieval artifacts beneath the emission, which would penalize the retrieved enhancements when expanding the matched filter spectral window of application. Although the use of Combo-MF should take into account the surface composition beneath potential methane emissions, it generally improves the detection capability. Note that the atmospheric elements that appear in the true-color image (top-left) are mainly related to pollutants following the Mie scattering, which does not substantially affect radiance at SWIR wavelengths (Roger et al., 2023).

### 3.2.3 Retrieval performance in a coal mining site using PRISMA data

Combo-MF is well-suited for regions with pronounced heterogeneity because there are usually more retrieval artifacts that can be removed. In order to demonstrate this, we conduct a comprehensive study in a PRISMA dataset from a coal mining site in Shanxi (China), which can be considered heterogeneous. In Figure 11, we illustrate in the central panel 15 plumes detected in the area with the automatic detection algorithm (see section 2), which are pointed out with red arrows, over a high-resolution image of the area. These plumes originate from potential emission sources and approximately align with the wind direction, which is extracted from GEOS-FP. Therefore, we validate the existence of these emissions. At the edges of the central panel, we observe four blocks associated with plumes (labeled $a - d$), displayed as an example of the detection process across the scene. In these blocks, we show the zoomed-in view of the emission source (gray) and of the output masks derived from the automatic detection algorithm used in both the 2300-MF (blue) and Combo-MF (green) retrievals. While the source in $a$ is associated with a drainage station, in $b - d$ we observe emissions emanating from venting shafts. In addition, the masked clusters associated with the plumes, framed in red, are illustrated along with their surroundings. We observe that the Combo-MF mask discards an important fraction of those clusters from the 2300-MF mask related to retrieval artifacts coming from surface structures such as roads or facilities. The clutter penalization from Combo-MF also reduces other pixel retrieved enhancements, resulting in a lower number of clusters related to background noise. However, clutter emergence does not always yields $\Delta XCH_4$ values in SWIR-MF greater than those in 2300-MF and therefore they cannot be removed according to Eq. 9. Instead, these values are scaled, which contributes to the appeareance of undesired clusters. For instance, right below the plume in $d$, a small cluster that is not present in 2300-MF appears in Combo-MF. In addition, the use of a more demanding target spectrum in Combo-MF can lead to the underestimation of weak plume pixels such as those from the plume tail. These pixels can be confused by background noise even after applying the scaling factor $f$. Therefore, plume clusters such as in $c$

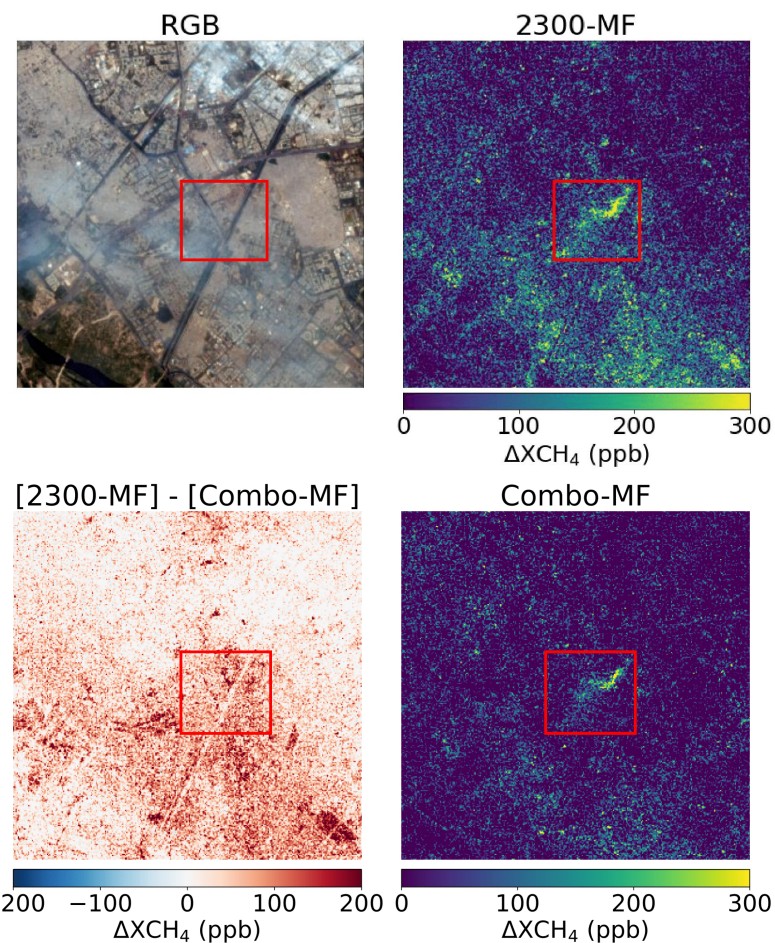

**Figure 10.** True color image (top-left), 2300-MF (top-right) and Combo-MF (bottom-right) retrievals, and the difference between them (bottom-left) from an EnMAP dataset from Delhi (India) showing an emission from the Gazhipur landfill (framed in red).

can present a more reduced shape than in 2300-MF. Altogether, we observe that there is a lower number of clusters around the plume when using Combo-MF, which facilitates plume detection.

## 4    Summary and conclusions

380    In this work, we propose a new matched filter-based procedure that attenuates or removes retrieval artifacts and also reduces the positive values of background noise. We investigated the retrieval performance in real observations of spectral imagers on satellites and airplanes, using real and simulated plumes, and found a significant increase in $CH_4$ emission detection. First, we implemented simulated $CH_4$ plumes in different PRISMA datasets and conducted a visual comparison. Second, we quantified

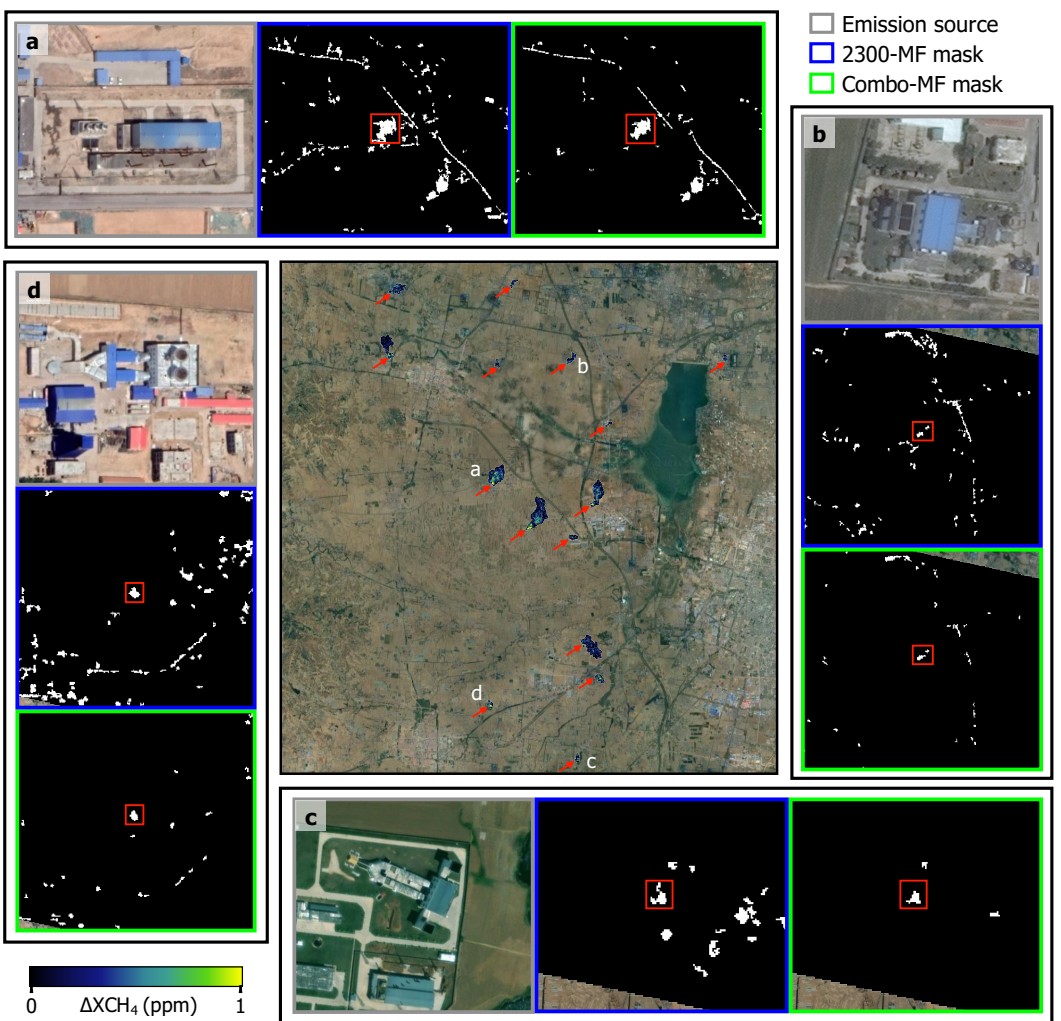

**Figure 11.** Plumes (pointed out with red arrows) detected with the automatic detection algorithm using a PRISMA dataset from a coal mine site in Shanxi (China), overlaid on a © Google Earth high-resolution image (central panel). At the edges, four blocks associated to plumes selected as examples (labeled $a - d$) display a zoomed-in view of the emission source (gray) and of the output masks from the automatic detection algorithm. The masks shows the cluster related to the plume (framed in red) and their surroundings for the 2300-MF (blue) and Combo-MF (green) retrievals.

the synthetic plumes implemented using masks derived from different retrievals. Third, we assessed the interference of $H_2O$

385   and $CO_2$ in $CH_4$ retrievals by means of simulations applied in PRISMA and EnMAP datasets. Finally, we evaluated retrievals containing $CH_4$ emissions from real AVIRIS-NG, EnMAP, and PRISMA data to compare the proposed procedure to already existing procedures using an automatic detection algorithm and visual inspection.

Our analysis reveals that Combo-MF effectively attenuates or practically removes an important fraction of retrieval artifacts while maintaining plume $\Delta XCH_4$ values at levels comparable to those obtained with 2300-MF. In addition, plume masking resulting from Combo-MF is less affected by background noise for small flux rate values ($Q < 1000$ kg/h), translating to a lower sensitivity to clutter. Nevertheless, emissions over surface structures linked to artifacts can be considerably attenuated in the Combo-MF retrieval due to the penalization of the underlying surface, as seen in the Gazhipur landfill case. Moreover, local enhancements of $H_2O$ or $CO_2$ interfering with $CH_4$ emissions, as observed in inefficient flaring cases, can attenuate the plume enhancement in the $\Delta XCH_4$ maps. On the other hand, atmospheric $H_2O$ and $CO_2$ concentrations have a negligible impact on the retrievals because of the homogeneous distribution of these trace gases across the retrieved area. We also evaluated the performance of MAG1C using the 1000-2500 nm spectral range, observing a reduction in the number of retrieval artifacts compared to the default 2122-2488 nm window. Despite this reduced number and the lower background noise, it generates plumes with higher $\Delta XCH_4$ values and still exhibits a greater number of artifacts compared to Combo-MF retrievals. Therefore, Combo-MF can be considered an optimal trade-off between background noise and retrieval artifact reduction, generally leading to an improved plume detection capability. This is illustrated by a comprehensive analysis in a PRISMA dataset, where the output masks from an automatic detection algorithm show an important reduction in the number of clusters not related to $CH_4$ emissions.

*Data availability.* Data will be made available on request.

*Author contributions.* **Javier Roger**: Conceptualization, Methodology, Formal analysis, Investigation, Writing – original draft, Writing – review & editing. **Luis Guanter**: Conceptualization, Resources, Writing – review & editing, Supervision. **Javier Gorroño**: Conceptualization, Methodology, Writing – review & editing. **Itziar Irakulis-Loitxate**: Formal Analysis, Resources, Writing – review & editing.

*Competing interests.* The authors declare that they have no conflict of interest.

*Acknowledgements.* The authors thank the Italian Space Agency, the DLR Space Agency, and, JPL team for the PRISMA, EnMAP, and AVIRIS-NG data used in this work, respectively. We are grateful to Daniel J. Varon for the WRF-LES modeled plumes used in this study. Authors Javier Roger, Javier Gorroño, and Luis Guanter received funding from ESA contract 4000134929.

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
