# Peer review of "Exploiting the entire near-infrared spectral range to improve the detection of methane plumes with high-resolution imaging spectrometers"

_Atmospheric Measurement Techniques, 2023_

## Referee Comment (RC1)

Review on "Exploiting the entire near-infrared spectral range to improve the detection of methane plumes with high-resolution imaging spectrometers".

The authors present an augmented version of the matched filter retrieval. The matched filter is employed in many studies for the identification and quantification of methane enhancements in hyperspectral data, commonly from airborne instruments in a top-down viewing geometry. Studies generally exploit methane absorption features in the shortwave infrared (SWIR) spectral range around the 2300 nm range for the algorithm. Yet, since the surface albedo may vary strongly in the SWIR, surface reflectance structures can align with the absorption features of methane spectrally and thus, cause artifacts in the produced image. These artifacts pose challenges to plume detection algorithms. Many state-of-the-art instruments observe wider spectral intervals in the SWIR. The authors present a technique to exploit the extended range between 1000 and 2500 nm to reduce image artifacts due to background heterogeneity. They evaluate the techniques performance on simulated and real data and compare it to similar retrieval adaptations from previous publications.

Monitoring anthropogenic methane emissions is an integral element of human efforts to mitigate climate change and has received particular attention in recent years. The matched filter retrieval is widely used in the scientific community due to its simplicity and efficiency. Retrieval artifacts have been identified as a major source of errors in the past, thus the findings of this paper are timely and scientifically significant. The performance of the adapted matched filter technique is impressive and achieves the goal of improving signal identification. Furthermore, it is applicable to many ongoing and upcoming satellite imaging instruments.
The applied methods are rigorous and valid, and the experiments support the findings. Yet, a quantification of some results is necessary to put them in context regarding applicability. Also, some open questions remain to be addressed in the review process.
The overall presentation quality is acceptable but should be improved during the review process. Some paragraphs are structured confusing, and findings are introduced before evidence of any kind was presented. Many paragraphs include colloquial language and need to be worded more concise.

Please find detailed comments below.

Major comments:

0. Abstract:
   a. Needs to be rephrased according to review changes.
1. Introduction
2. Materials and methods
   a. L111: The potential of the matched filter to use spectral information from distant bands to remove artifacts is (a) a conceptual difference to physical-based methods, which cannot do this, and (b) the main reason why the method presented in this paper works. Please dedicate more space to explain the idea and how information is communicated from the newly integrated bands via the covariance matrix of the scene.
   b. L122ff: The following paragraphs use results of the paper without stating this explicitly, such that they appear as claims without evidence. I believe the authors try to explain the behaviour of their adapted matched filter, but without the evidence/experience of the result section, these paragraphs are confusing. I try to go through them in detail.
   c. L122: "Although SWIR-MF reduces retrieval artifacts [..]" – has not been shown yet.

d. L129: "[…], which leads to a lower weight of the background variance direction." – I do not understand this, what is a weight in this context and how does it affect the retrieval performance? Are you referencing to the $C^{-1}t$ – vector?

e. L130: "This difficults the discrimination between plume pixels and background pixels, which can lead to enhanced background clutter and plume pixel attenuation." – I don't understand in detail why a smaller correlation between distant bands removes information from the filter. The information of the closer bands is still included, and also figure 6 shows that the SWIR-MF has a smaller variability.

f. Figure 4: Please add axis labels.

g. L140: How do you arrive at the conclusion that co-emitted carbon dioxide (a) affects the retrieval and (b) decreases the methane columns? Might this be mitigated by an exclusion of the carbon dioxide feature bands from the matched filter?

h. L142ff: I found the following paragraph very hard to understand.
As I understand it, the SWIR-MF produces overall smaller enhancements than the 2300-MF. This is especially true for 2300-MF artifacts and clutter regions. There are exceptions, in which the SWIR-MF will produce enhancements greater than the 2300-MF, which can be attributed to spurious correlations in the additional 1700 nm methane band. These exceptions are dealt with by simply replacing them with the 2300-MF clutter retrieval values, which results in the COMBO-MF. In order to keep the variances of the retrievals comparable, you introduce the scaling factor $f$. This procedure removes artifact from both retrievals and large clutter values of the SWIR-MF, thus increasing the contrast between background and plume pixels.
If my understanding is correct, consider rewording the paragraph along these lines. Also, I strongly suggest writing down a formula to make the procedure clear ($\alpha = \Delta CH4$):
$$\alpha_{combo} = \begin{matrix} \alpha_{swir} * f & if \ \alpha_{swir} < \alpha_{2300} \\ \alpha_{2300} & if \ \alpha_{swir} \geq \alpha_{2300} \end{matrix}$$

i. L153: What is a standard deviation here? Please give a definition.

j. L151: If you use the SWIR-MF as the more trustworthy retrieval and it has a smaller variance, why don't you scale the 2300-MF values before including them?

k. Figure 5: I believe the conditional comparisons are exchanged. Please include a formula for the technique, then I would suggest removing this figure.

l. The condition of the COMBO-MF will remove plume pixels which are detected in the 2300-MF but remain undetected in the SWIR-MF. Can you comment on if you could observe this behaviour, and if it might cause systematic biases for emissions estimates?

m. L206: The masking criteria is a bit strange. Why do you include your knowledge about the synthetic plume here? This removes partly the effect of artifact suppression on the plume mask, right?

3. Results
   a. L256: As I understand it, the GF5-02 dataset includes a real CH4 plume. If that is the case, move it to section 3.2.
   b. L56ff: Is the whole argument in section 2 about the matched filter underestimating CH4 in the case of co-emitted CO2 based on this result? If yes, consider supporting it further with e.g., the retrieval of synthetic co-emitted plumes. It feels you highlight this finding a lot, therefore it needs more evidence.
   c. L288ff: I agree that the noise suppression works fine in this example, but the visual plume detection is an insufficient argument here. It would be much more robust if you used a plume detection algorithm and attributed the identified plumes to coal mine

> ventilation shafts on the ground. Do you have the locations of the shafts, or how do you know that the identified plumes are indeed methane emissions?

4. Summary and conclusions
   a. Needs to be rephrased according to review changes.

Minor comments:

0. Abstract:
   a. You employ a lot of passive speech, consider wording your sentences in an active way. This way, the reader knows who/what performs a given task.
1. Introduction
   a. l50ff: This paragraph motivates the whole publication, please add some references which highlight the necessity, e.g., https://doi.org/10.1016/j.rse.2018.06.018.
   b. L36ff: Please make clear to which instruments you compare the hyperspectral imagers – they do not have a high spectral resolution objectively.
   c. L52: "raw" matched filter is not defined. Description required.
   d. Figure 1: "[…] resampled to 2 nm spectral *resolution*". Also put "wavelength" on x-axis as long as lambda has not been introduced.
   e. Figure 2: What does "detected" mean in the context of this figure? How can you be sure all other detections are artifacts?
2. Materials and methods
   a. L65ff: Unit absorption spectra generation is not well described. Clearly state that you collect absorption cross-sections from HITRAN and use them in a simple lambert-beer transmission model, which accounts for observation and solar angles.
   b. L71: As far as I know, HITRAN does not calculate radiative transfer, therefore the scattering-sentence is confusing. Furthermore, you cannot state that scattering is negligible in the SWIR. Aerosol impact on retrievals is still ongoing research, and clouds most definitely have an impact on SWIR RT. Please formulate your assumption of a pristine atmosphere to back this simplified radiative transfer model.
   c. L90: This sentence belongs to the unit absorption spectrum paragraph.
   d. L65-L98: These paragraphs give a broader overview of the matched filter, bus are not sufficient to explain the method for an untrained reader. It is out of the scope of this paper to explain the matched filter in detail but add an introductory sentence in the beginning which redirects to explanatory literature would be appreciated.
   e. L111: "should be" is overused in this paper. It undermines the reliability of the findings, consider wording your findings more confidant if you are, or remove the sentences if it is speculation.
   f. Figure 3: Please mention in the caption that this is a 2-band matched filter.
   g. L183: Add a sentence on how the plumes are implemented in the dataset.
   h. L196: $U_{eff}$ depends logarithmic on $U_{10}$
   i. L203: What is a standard deviation in this context?
   j. L215ff: Please mention which of the datasets include real methane enhancements and which of the are free from real sources, such that they may be used for the synthetic studies.
   k. L220: "Moreover, [..]" – add references to figures 3 and 4.
3. Results
   a. Figure 7: A zoom to the plume would help to show the effect on plume shape. Optional comment.

b.  Figure 8: The distributions of the data points seem to be highly asymmetrical for smaller fluxes, since the uncertainty ranges reach to the negative. Could you make a boxplot for each flux?

c.  L244: "lower" – please provide numbers

d.  L266: You already commented on the "upper limit" in the text and in figure 10. If this is only a discussion about the colorbar-ranges, please limit it to the figure caption. If you have introduced an upper limit to one of the methods, this needs much more explanation.

e.  L275: The last sentence seems to be connected to the unit absorption spectrum generation, please clarify this.

f.  L284: "somewhat attenuated" → quantify, might this be due to my comment 2.l ?

4.  Summary and conclusions

a.  L299: explicitly name the instrument from which the data was taken.

b.  L300: "different masking methods" sounds like you did a performance analysis. Make clear that results are either based on a thresholding approach or visual identification.

c.  L316: You could highlight that the improvement is especially prominent for small fluxes, which are typically hard to identify.

Technical corrections:

0.  Abstract:
1.  Introduction

a.  L31: "[..] come*s* from [..]" – missing 's'

b.  L37ff: "we find" is colloquial English, please use scientific language.

c.  L45ff: whole paragraph is colloquial English.

d.  L46: "[..] its interactions and the media *that travels* through." – grammatical error.

e.  L58: "elaborated" – use present tense, e.g. "In this work, we present […]"

2.  Materials and methods

a.  L64: "methods" not capitalized in heading

b.  L77: introduce vector notation from beginning, please write "spectral mean vector" at the first occurrence and type all vectorial quantities bold.

c.  L98: "in a per-column basis." → "for each along-track column separately."

d.  L101: citation order from oldest to youngest paper.

e.  L115: *a* scatter plot

f.  L116: a*n* EnMAP data set

g.  L128: scattered → wide-spread

h.  L130: difficults → complicates

i.  L133: not removed → additional

j.  L158: remove "as" before "COMBO-MF"

k.  L191: should not → will not

l.  L210: s*u*rroundings

m.  L279: we can observe → shows

n.  L280: "here there are" is colloquial, please rephrase

o.  L283: "the positive values" – unnecessary *s*

p.  L284: "migh" → might

3.  Results
4.  Summary and conclusions

a.  L304: not-simulated → real

---

## Author Comment (AC1)

**SWIR-MF**

$\Delta XCH_4 (1)|_{SWIR-MF}$ : $\Delta XCH_4|_{SWIR-MF} > \Delta XCH_4|_{2300-MF}$

$$f = \frac{\sigma_{2300-MF}}{\sigma_{SWIR-MF}}$$

$\Delta XCH_4 (1)|_{Combo-MF} = f \cdot \Delta XCH_4 (1)|_{SWIR-MF}$

$\Delta XCH_4 (2)|_{SWIR-MF}$ : $\Delta XCH_4|_{SWIR-MF} \leq \Delta XCH_4|_{2300-MF}$

$\Delta XCH_4 (2)|_{Combo-MF} = \Delta XCH_4 (2)|_{2300-MF}$

**Combo-MF**

---

## Author Comment (AC2)

SWIR-MF

$\Delta XCH_4 (1)|_{SWIR-MF} : \Delta XCH_4 |_{SWIR-MF} < \Delta XCH_4|_{2300-MF}$

$$f = \frac{\sigma_{2300-MF}}{\sigma_{SWIR-MF}}$$

$\Delta XCH_4 (1)|_{Combo-MF} = f \cdot \Delta XCH_4 (1)|_{SWIR-MF}$

$\Delta XCH_4 (2)|_{SWIR-MF} : \Delta XCH_4|_{SWIR-MF} \geq \Delta XCH_4|_{2300-MF}$

$\Delta XCH_4 (2)|_{Combo-MF} = \Delta XCH_4 (2)|_{2300-MF}$

Combo-MF

---

## Author Response (AR1)

Dear Reviewers,

Thank you for the careful and comprehensive review of the manuscript. The main changes performed on the manuscript have been:

- The clarification of the matched filter concept.

- The use of an automatic detection algorithm to detect methane plumes.

- The quantification tests analysis is only made to assess the influence of background noise in Combo-MF.

- The use of simulations to assess the influence of the overlap of CO2 and H2O local enhancements with CH4 plumes.

In addition to those major changes, a number of corrections and clarifications have been made throughout the text regarding other major and minor concerns. Please, find below point-by-point responses (in blue) to your comments and suggestions.

Kind regards,

Javier Roger, on behalf of the authors

**Reviewer 1**

Review on "Exploiting the entire near-infrared spectral range to improve the detection of methane plumes with high-resolution imaging spectrometers".

The authors present an augmented version of the matched filter retrieval. The matched filter is employed in many studies for the identification and quantification of methane enhancements in hyperspectral data, commonly from airborne instruments in a top-down viewing geometry. Studies generally exploit methane absorption features in the shortwave infrared (SWIR) spectral range around the 2300 nm range for the algorithm. Yet, since the surface albedo may vary strongly in the SWIR, surface reflectance structures can align with the absorption features of methane spectrally and thus, cause artifacts in the produced image. These artifacts pose challenges to plume detection algorithms. Many state-of-the-art instruments observe wider spectral intervals in the SWIR. The authors present a technique to exploit the extended range between 1000 and 2500 nm to reduce image artifacts due to background heterogeneity. They evaluate the techniques performance on simulated and real data and compare it to similar retrieval adaptations from previous publications.

Monitoring anthropogenic methane emissions is an integral element of human efforts to mitigate climate change and has received particular attention in recent years. The matched filter retrieval is widely used in the scientific community due to its simplicity and efficiency. Retrieval artifacts have been identified as a major source of errors in the past, thus the findings of this paper are timely and scientifically significant. The performance of the adapted matched filter technique is impressive and achieves the goal of improving signal identification. Furthermore, it is applicable to many ongoing and upcoming satellite imaging instruments.

The applied methods are rigorous and valid, and the experiments support the findings. Yet, a quantification of some results is necessary to put them in context regarding applicability. Also, some open questions remain to be addressed in the review process.

The overall presentation quality is acceptable but should be improved during the review process.

Some paragraphs are structured confusing, and findings are introduced before evidence of any kind was presented. Many paragraphs include colloquial language and need to be worded more concise.

Please find detailed comments below.

**Major comments:**

**0. Abstract:**

a. Needs to be rephrased according to review changes.

Corrected.

**1. Introduction**

**2. Materials and methods**

a. L111: The potential of the matched filter to use spectral information from distant bands to remove artifacts is (a) a conceptual difference to physical-based methods, which cannot do this, and (b) the main reason why the method presented in this paper works. Please dedicate more space to explain the idea and how information is communicated from the newly integrated bands via the covariance matrix of the scene.

In new section 2.2. (called *Combo-MF* – added to improve the structure of the work), we explain why physically-based method are not well-suited for using a wider spectral range of application. Moreover, we add that there is a very low added computational processing time involved when using SWIR-MF instead of 2300-MF.

In L99-141 from the previous version, an analysis was made in order to explain how expanding the spectral window can help to attenuate/remove retrieval artifacts. However, after reviews, we considered that a more comprehensive analysis was needed. Therefore, we remove this fragment from the manuscript and write an improved version. The main changes applied are:

- We include the greater spectral sampling of the 2300 nm methane absorption window in comparison to the 1700 nm window as an additional reason to explain why the 2300 nm window is typically chosen.

- We remove the spectral decorrelation between distant bands as an important clutter source in the methane retrieval. We found that in some datasets from different missions, there was not a remarkable increasing decorrelation between distant bands, but clutter noise still appeared. Instead, we include the higher sensitivity of the matched-filter due to a more demanding spectrum to match as a more important clutter source when expanding the window. We added: 'This increased sensitivity could also capture undesired variations from the modeled spectrum due to factors unrelated to methane, such as atypical albedo values or instrument noise, potentially resulting in clutter noise'. Therefore, we remove the old Figure 4.

- In the previous version, we used the concept of the matched-filter direction as a balance between the reference spectrum direction and the minimum background variance direction, which was extracted from Eismann et al., (2012). However, this concept was visually difficult to illustrate and

increased the complexity of the explanation. Therefore, we remove this concept (and the old Figure 3) from the manuscript and use a different perspective. We examine the functionality of the matched-filter using only 2 bands with the new Figure 3. We describe how the matched-filter performs a real-like absorption, an absorption that perfectly follows the model (new Eq.3), and a retrieval artifact. This explanation clarifies how the matched-filter works and why expanding the window could lead to the attenuation of the retrieval artifacts.

- We also include the new Figure 4, which shows the retrieval performance when expanding the window. This clarifies how information is communicated from the newly integrated bands via the covariance matrix of the scene. Moreover, it helps to show the disadvantages from using SWIR-MF.

- We justify the removal of bands related to strong H2O absorption features around 1400 nm and 1900 nm because of the very low radiance of these bands can negatively impact the retrieved enhancement values.

b. L122ff: The following paragraphs use results of the paper without stating this explicitly, such that they appear as claims without evidence. I believe the authors try to explain the behaviour of their adapted matched filter, but without the evidence/experience of the result section, these paragraphs are confusing. I try to go through them in detail.

We apply a comprehensive revision across the whole text in order to solve this issue.

Note that in Section 2.1. we included the new Figure 4, which analyzes the impact on the retrieved values when expanding the window in a PRISMA dataset. Although this figure can be considered as a result and therefore should be included in the *Results* section, we believe that we need this figure in the *Materials and Methods* section to clearly explain the new methodology. Something similar occurs with the new Figure 3 and 5.

c. L122: "Although SWIR-MF reduces retrieval artifacts [..]" – has not been shown yet.

We remove this sentence and only confirm the retrieval artifact reduction since the *Results* section.

d. L129: "[…], which leads to a lower weight of the background variance direction." – I do not understand this, what is a weight in this context and how does it affect the retrieval performance? Are you referencing to the C-1 t – vector?

We remove this sentence. As replied in comment 2.a., we discard the matched-filter direction concept from Eismann et al., (2012).

e. L130: "This difficults the discrimination between plume pixels and background pixels, which can lead to enhanced background clutter and plume pixel attenuation." – I don't understand in detail why a smaller correlation between distant bands removes information from the filter. The information of the closer bands is still included, and also figure 6 shows that the SWIR-MF has a smaller variability.

We remove this sentence. As replied in the comment 2.a., we discard the decorrelation between distant bands as an important error source in the retrieved values when expanding the window.

f. Figure 4: Please add axis labels.

As replied in comment 2.a., we remove the old Figure 4. In addition, we check that the labels of the axes of the rest of the figures are added and correct.

g. L140: How do you arrive at the conclusion that co-emitted carbon dioxide (a) affects the retrieval and (b) decreases the methane columns? Might this be mitigated by an exclusion of the carbon dioxide feature bands from the matched filter?

We remove these statements from the *Materials and methods* section. In addition, we add a study in section 3.1. based on simulations of overlapping local enhancements of $CO_2$ and $H_2O$ over a methane plume and confirmed (a) and (b). In this study, we also show that the attenuation due to the overlapped enhancements of $CO_2$ in the 2300-MF retrieval is almost negligible due to the little $CO_2$ absorption in the 2300 nm window. This demonstrates that the exclusion of $CO_2$ features bands can mitigate the impact on methane emission retrieved values. We include this concept in the text.

h. L142ff: I found the following paragraph very hard to understand. As I understand it, the SWIR-MF produces overall smaller enhancements than the 2300- MF. This is especially true for 2300-MF artifacts and clutter regions. There are exceptions, in which the SWIR-MF will produce enhancements greater than the 2300-MF, which can be attributed to spurious correlations in the additional 1700 nm methane band. These exceptions are dealt with by simply replacing them with the 2300-MF clutter retrieval values, which results in the COMBO-MF. In order to keep the variances of the retrievals comparable, you introduce the scaling factor f. This procedure removes artifact from both retrievals and large clutter values of the SWIR-MF, thus increasing the contrast between background and plume pixels. If my understanding is correct, consider rewording the paragraph along these lines. Also, I strongly suggest writing down a formula to make the procedure clear ( $= \Delta CH4$): $combo = swir * f \, if \; swir < 2300 \quad 2300 \, if \; swir \geq 2300$

Corrected.

i. L153: What is a standard deviation here? Please give a definition.

Corrected.

j. L151: If you use the SWIR-MF as the more trustworthy retrieval and it has a smaller variance, why don't you scale the 2300-MF values before including them?

The included 2300-MF values are typically negative, as we can see in the histograms from the new Figure 5. A higher or lower negative value will have no impact on detection because the lower limit for representation always is the zero value. However, although we did such modification, we would scale these values to make them comparable to the ones of 2300-MF. This is done because in SWIR-MF we find an underestimation of the retrieved values (see new Figure 4). Although the SWIR-MF is more trustworthy regarding retrieval artifacts, the modifications in Combo-MF are done to attenuate the emergence of clutter and the underestimation of the retrieved values.

k. Figure 5: I believe the conditional comparisons are exchanged. Please include a formula for the technique, then I would suggest removing this figure.

Corrected. We remove the figure and include a formula.

l. The condition of the COMBO-MF will remove plume pixels which are detected in the 2300-MF but remain undetected in the SWIR-MF. Can you comment on if you could observe this behaviour, and if it might cause systematic biases for emissions estimates?

We do observe this behavior. We include this situation in the new Section 2.2: *Combo-MF*, which is included to better separate the information in the *Materials and Methods* part. Moreover, this situation is explained in a real case (3.2.2.), where a landfill emission from Delhi was detected with EnMAP.

m. L206: The masking criteria is a bit strange. Why do you include your knowledge about the synthetic plume here? This removes partly the effect of artifact suppression on the plume mask, right?

The information about the synthetic plume partly removes this effect because the attenuation of artifacts makes them less prone to be included in the masks. After this consideration, we realized that this masking criterion could not be applied to real data. Any conclusions from our quantification tests would not be useful for practical applications.
Regarding artifact suppression due to the information of the synthetic plume, we can approximately isolate background noise as the only disturbing factor. Thus, we carry out these tests in order to analyze how background noise can affect masking and therefore quantification. We include this explanation in the quantification section (new 2.4. section).

**3. Results**

a. L256: As I understand it, the GF5-02 dataset includes a real CH4 plume. If that is the case, move it to section 3.2.

We remove this figure. The comment 2.g motivated us to use simulations to study the interference of other trace gases with methane plumes. Compared to these new results, we consider that the old Figure 9 was not a significant contribution to illustrate this interference, as the attenuation was not strong enough to derive any solid conclusions.

b. L56ff: Is the whole argument in section 2 about the matched filter underestimating CH4 in the case of co-emitted CO2 based on this result? If yes, consider supporting it further with e.g., the retrieval of synthetic co-emitted plumes. It feels you highlight this finding a lot, therefore it needs more evidence.

Corrected. See reply to comment 3.a.

c. L288ff: I agree that the noise suppression works fine in this example, but the visual plume detection is an insufficient argument here. It would be much more robust if you used a plume detection algorithm and attributed the identified plumes to coal mine ventilation shafts on the ground. Do you have the locations of the shafts, or how do you know that the identified plumes are indeed methane emissions?

We apply an automatic detection algorithm (see explanation in new section 2.4.) that outputs a mask showing a selection of clusters that are related to potential plumes. This algorithm has been applied to the Combo-MF and 2300-MF retrievals that were illustrated in the old Figure 12. In this manner, we introduce a systematic methodology to evaluate the methane plume detection capability. Therefore, we remove the old Figure 12 and add the new Figure 11. Moreover, we modify section 3.2.3. to modify the explanation according to the new results.

In addition, we show a zoomed-in view of 4 plume sources with high-resolution images from Google Earth: 3 venting shafts and 1 drainage station. Plume detection is confirmed following the

steps indicated in the new section 2.4., where it is indicated how we can identify a plume. Note that this information about detection was not present in the previous version of the manuscript.

**4. Summary and conclusions**

a. Needs to be rephrased according to review changes.

Corrected.

**Minor comments:**

**0. Abstract:**

a. You employ a lot of passive speech, consider wording your sentences in an active way. This way, the reader knows who/what performs a given task.

Corrected. We reduce the presence of passive speech.

**1. Introduction**

a. l50ff: This paragraph motivates the whole publication, please add some references which highlight the necessity, e.g., https://doi.org/10.1016/j.rse.2018.06.018.

Corrected.

b. L36ff: Please make clear to which instruments you compare the hyperspectral imagers – they do not have a high spectral resolution objectively.

We remove the adjective 'high' to avoid adding the comparison. We are only interested in stating that imaging spectrometers are able to resolve a large range of point-sources.

c. L52: "raw" matched filter is not defined. Description required.

We remove 'raw' and only keep 'matched-filter' as in Ayasse et al., (2018).

d. Figure 1: "[…] resampled to 2 nm spectral *resolution*". Also put "wavelength" on x-axis as long as lambda has not been introduced.

Corrected.

e. Figure 2: What does "detected" mean in the context of this figure? How can you be sure all other detections are artifacts?

In Section 2.4 we add the steps we follow to confirm the detection of a plume. Here we do mention that the emission from Figure 2 was the only detected plume following these steps.

**2. Materials and methods**

a. L65ff: Unit absorption spectra generation is not well described. Clearly state that you collect absorption cross-sections from HITRAN and use them in a simple lambert-beer transmission model, which accounts for observation and solar angles.

We used MODTRAN instead of HITRAN. Sorry for the inconvenience. We correct this.

b. L71: As far as I know, HITRAN does not calculate radiative transfer, therefore the scattering-sentence is confusing. Furthermore, you cannot state that scattering is negligible in the SWIR. Aerosol impact on retrievals is still ongoing research, and clouds most definitely have an impact on SWIR RT. Please formulate your assumption of a pristine atmosphere to back this simplified radiative transfer model.

See reply to comment 2.a.

Regarding the scattering in the SWIR… corrected.

c. L90: This sentence belongs to the unit absorption spectrum paragraph.

Corrected.

d. L65-L98: These paragraphs give a broader overview of the matched filter, bus are not sufficient to explain the method for an untrained reader. It is out of the scope of this paper to explain the matched filter in detail but add an introductory sentence in the beginning which redirects to explanatory literature would be appreciated.

As replied in the comment 2.a from major concerns, we clarify the matched filter concept. Besides, several references related to the matched filter are already included in the manuscript. Therefore, we decide not to include an introductory sentence redirecting to explanatory literature.

e. L111: "should be" is overused in this paper. It undermines the reliability of the findings, consider wording your findings more confidant if you are, or remove the sentences if it is speculation.

Corrected. Note that 'should be' is only used in Section 2.1. to introduce the hypothesis of retrieval invariability when adding atmospheric $H2O$ or $CO2$ enhancements. However, this hypothesis is confirmed in the *Results* section.

f. Figure 3: Please mention in the caption that this is a 2-band matched filter.

As replied in the comment 2.a from major concerns, we remove the old Figure 3. However, the new Figure 3 also exhibits a scatter plot with 2-band matched filter values. We include this in the caption.

g. L183: Add a sentence on how the plumes are implemented in the dataset.

Added.

h. L196: U_eff depends logarithmic on U_10

We wrote 'linearly' because we used the Ueff from Guanter et al. (2021):

Ueff = 0.34 * U10 + 0.44

This effective wind speed was adapted to the PRISMA instrument and can also be used for EnMAP (Roger et al., 2023). We use this calibration to obtain the results that are shown in the new Figure 7 because we use PRISMA datasets. We add this in the text.

i. L203: What is a standard deviation in this context?

Corrected. We clarify the concept of standard deviation for cases that present ambiguity.

Note that we removed the list of steps for masking. We introduce the masking process as a part of an automatic detection algorithm (new section 2.4.), motivated by the major comment 3.c.

j. L215ff: Please mention which of the datasets include real methane enhancements and which of the are free from real sources, such that they may be used for the synthetic studies.

 Corrected.

k. L220: "Moreover, [..]" – add references to figures 3 and 4.

Corrected.

**3. Results**

a. Figure 7: A zoom to the plume would help to show the effect on plume shape. Optional comment.

We applied this view in order to see how the surroundings are modified for Combo-MF in comparison to 2300-MF. However, in the new Figure 11, we can observe the clusters related to plumes for both retrievals and their different shapes.

b. Figure 8: The distributions of the data points seem to be highly asymmetrical for smaller fluxes, since the uncertainty ranges reach to the negative. Could you make a boxplot for each flux?

We consider that a box plot is not necessary. Regarding the old Figure 8 (new Figure 7), we can conclude that uncertainty levels depend on the surface type, the more restrictive masking from Mix-MF is less sensitive to background noise in comparison to the one from 2300-MF, and both quantification procedures converge at relatively high flux rates values (>1000 kg/h).

Note that the masking process involves a previous median filter. Negative values from the retrieval can be included in the plume mask because of this reason. In some cases, the resulting IME could be negative and therefore lead to negative flux rate values.

c. L244: "lower" – please provide numbers

Because the masking process cannot be applied to a real case due to the use of the synthetic plume information, we consider that a quantitative analysis for the uncertainty is not needed. Therefore, we keep a qualitative analysis.

d. L266: You already commented on the "upper limit" in the text and in figure 10. If this is only a discussion about the colorbar-ranges, please limit it to the figure caption. If you have introduced an upper limit to one of the methods, this needs much more explanation.

Here we made a mistake. In the old Figure 10 (new Figure 9) the same upper limit for representation was used for all the retrievals. We correct this.

e. L275: The last sentence seems to be connected to the unit absorption spectrum generation, please clarify this.

In Section 2.1., we mentioned: 'Note that k_CH4 in satellite-based missions is calculated considering the integration of CH4 over an 8 km high column such as in Thompson et al. (2016), while in airborne missions is calculated over the specific flight height'. Therefore, in this section (3.2.1.), we note that we use 2.48 km because it is the scene average sensor altitude at acquisition time from an airborne instrument. We clarify this in the text.

f. L284: "somewhat attenuated" → quantify, might this be due to my comment 2.l ?

In order to better show the attenuation of landfills, we remove the old Figure 11 and include the new Figure 10. Here only the Gazhipur landfill and its related emission are shown. The removal of clutter can still be observed. In the bottom-left panel we include the difference between the 2300-MF and the Combo-MF retrievals. We use this panel to quantitatively illustrate the attenuation in Combo-MF.

We discard the quantification of the landfill emission because it is an area source and the quantification is different from point-sources (Maasakkers, 2022). Currently, we do not have the calibrations related for this kind of sources. Their calculation would involve a lot of work. Thus, we consider that adding the difference image in the bottom-left panel is a trade-off between showing quantitatively the attenuation and do not get involved in the calculation of a new calibrations.

Moreover, this indeed occurs according to the comment 2.l. We clarify this in this section (3.2.2.) and in section 2.2.

**4. Summary and conclusions**

a. L299: explicitly name the instrument from which the data was taken.

Corrected.

b. L300: "different masking methods" sounds like you did a performance analysis. Make clear that results are either based on a thresholding approach or visual identification.

We remove this and change the sentence to clarify that masks are deduced from different retrievals. Later on this section, we include the visual inspection and the automatic detection algorithm to identify plumes.

c. L316: You could highlight that the improvement is especially prominent for small fluxes, which are typically hard to identify.

The comment 2.m from major corrections, motivated us to only use this tests to study the influence of background noise on the masking process. However, we clarify that this influence is lower for small fluxes.

**Technical corrections:**

**0. Abstract:**

**1. Introduction**

a. L31: "[..] come*s* from [..]" – missing 's'

Corrected.

b. L37ff: "we find" is colloquial English, please use scientific language.

Corrected.

c. L45ff: whole paragraph is colloquial English.

Corrected.

d. L46: "[..] its interactions and the media *that travels* through." – grammatical error.

Corrected.

e. L58: "elaborated" – use present tense, e.g. "In this work, we present […]"

Corrected.

**2. Materials and methods**

a. L64: "methods" not capitalized in heading

Corrected.

b. L77: introduce vector notation from beginning, please write "spectral mean vector" at the first occurrence and type all vectorial quantities bold.

Corrected.

c. L98: "in a per-column basis." → "for each along-track column separately."

Corrected.

d. L101: citation order from oldest to youngest paper.

Corrected. Also done with the references to MAG1C of the new section 2.2.

e. L115: *a* scatter plot

Text changed.

f. L116: a*n* EnMAP data set

Text changed.

g. L128: scattered → wide-spread

Text changed.

h. L130: difficults → complicates

Text changed, but this change is done in other parts of the text.

i. L133: not removed → additional

Text changed.

j. L158: remove "as" before "COMBO-MF"

Corrected.

k. L191: should not → will not

We preserve should not. See reply to comment 2.e from minor comments.

l. L210: s*u*rroundings

Text changed.

m. L279: we can observe → shows

Corrected.

n. L280: "here there are" is colloquial, please rephrase

Text changed.

o. L283: "the positive values" – unnecessary *s*

Text changed.

p. L284: "migh" → might

Text changed.

**3. Results**

**4. Summary and conclusions**

a. L304: not-simulated → real

Text changed.

**Reviewer 2**

This paper presents a new matched filter algorithm for retrieval of methane plumes from point sources using the full SWIR spectrum. The idea is to minimize artifacts from surface features that may have spectral signatures in the 2.3 um methane band but may also have spectral features outside that band that would signal them as artifacts. This is a clever idea and well executed, and the results show significant improvements over standard matched filter retrievals. The text is wordy and there are many grammatical mistakes but these can be corrected with some careful editing. I support publication in AMT and have only a few comments for the authors' consideration below.

Line 23: methane is the second most important ANTHROPOGENIC greenhouse gas…

Corrected.

Line 68: alpha should be a column concentration.

Corrected. Additionally, we remove the alpha symbol to use just one CH4 column concentration enhancement symbol: ΔXCH4, which is later used on the study.

Figure 2, caption: detected by who?

This detection was made by the authors as part of the this study. We consider that it is not necessary to mention this in the caption. However, in Section 2.4 we add the steps we follow to confirm the detection of a plume. Here we do mention that the emission from Figure 2 was detected following these steps.

Line 72: saying that scattering is not relevant in the SWIR is a bit cavalier. Dust particles are efficient scatterers in that region of the spectrum.

Corrected.

Somewhere in Section 2.1: is there a significant computational performance penalty from using SWIR-MF compared to 2300-MF that would matter in practical applications?

In order to better structure the *Materials and Methods* section, we add the subsection called *Combo-MF* (2.2.). Here we include: 'Also note that the increased number of spectral bands does not substantially impact the computational processing time required when applying the matched-filter'. In fact, based on our experience for PRISMA and EnMAP datasets, SWIR-MF takes less than 10 additional seconds compared to 2300-MF.

Line 136: What about H2O features in the plume?

The atmospheric H2O concentration can generally be assumed homogeneous across the dataset area and therefore the H2O absorption features will be included in the mean value. Thus, deviations from the mean value cannot be related to the atmospheric H2O concentration. Simulations of atmospheric H2O have been implemented to confirm this hypothesis (see section 2.3. and 3.1.).

On the other hand, a study of a local overlap of H2O enhancements with a CH4 plume based on simulations is added in the new version of the manuscript (see section 2.3. and 3.1.) in order to analyze the impact of H2O features. We also include this study for the CO2 case. Here we observe that the local overlap of H2O and CO2 enhancements with a CH4 plume leads to an underestimation of the retrieved plume when expanding the matched-filter spectral window.

---

## Referee Report (RR1)

**Second review on "Exploiting the entire near-infrared spectral range to improve the detection of methane plumes with high-resolution imaging spectrometers"**

Major Comments:

None

Minor Comments:

Line 139: "Different original values, characterized by diverse deviations, could have also led to overestimated dXCH4 values. Therefore, deviations from the model will probably introduce biases in the retrieved values." – As I understand it, I wouldn't call this a bias. The calculated enhancements just include the noise of the underlying data, as is to be expected. In the case of a two-channel retrieval, this noise is high, but it reduces with increasing information content from more channels (up to a point, as you illustrate in Figure 4). I suggest rephrasing to something like "Natural enhancements scatter around their true value since the measurements are subject to noise, thus an overestimation would have been equally likely."

L156: "… which leads to a background noise reduction but also an underestimation of enhanced pixels." – You've shown that with your plot, no need to use "could".

L159: I think you should state here clearly that artifact reduction is a major selling point of the retrieval. You should refer that it is a result of your investigation - something along the lines of "New retrieval artifacts may appear when including the 1700 nm absorption window, but overall, the increased spectral interval mitigates false detections efficiently, as is shown in section 3."

L165: Since the 0-radiance pixels between 1800-1950 nm do not cause an increase in variability, I am not convinced that your explanation is correct. Does the retrieval noise shrink again when you exclude the water bands?
In any case, it is reasonable to exclude these intervals, they may cause trouble, and do not add value to the retrieval. But it seems to me more like the non-zero values below 1400 nm cause the rise in uncertainty, which is somehow interesting.
If you have strong evidence that it is the water bands (like the decrease of noise after exclusion), state it explicitly. If not, just note that you are removing the water bands for the abovementioned reasons and that the noise increase is due to the addition of the bright channels below 1400 nm.

L176: Since the SWIR range from 1000-2500 nm shows a higher noise, why didn't you choose the interval from 1500-2500 nm for the COMBO-MF? I guess the even broader interval might be more capable of reducing artifacts, even though it has a larger variability. If you chose it because of that, you should state it here to avoid confusion.

Caption Figure 8: Add the subscript CH4 to the Q in the caption. Also, you should mention in the caption that the constant enhancement of CO2 and H2O is only added to the plume pixels.

Figure 8+10: You might consider flipping the colorbar of the red-to-blue differences, it is more common to denote positive values with red and negative with blue.

L353: "Therefore, an appropriate use of Combo-MF should take into account the surface composition

beneath potential methane emissions." – This is hard to accomplish in reality, right? In any case, you can be more optimistic here. It is also possible that the COMBO-MF removes the influence of the air pollutants, which seem to significantly increase noise in the 2300-MF.

Figure 11: I think you should state here one time clearly that all shown plumes pass your plume masking procedure, and a-d are taken out as examples.

L380: I suggest adding a sentence which gives a broad overview, like "We investigated the retrieval performance in real observations of spectral imagers on satellites and airplanes, both using real and simulated plumes, and found a significant increase in the plume identification capability." The investigation of the retrieval using a broad spectrum of scenes and instruments was done thoroughly and detailed in this study, and should be highlighted in the beginning of the summary.

L390: You don't mention that you recommend he Mix-MF for emission quantification. You should mention it in the summary, and it fits when you point out the caveats of the COMBO-MF.

Technical corrections:

L156: make*s* the

L203: "Moreover, in Figure 5 we can see …" is wordy and somehow colloquial. I suggest rephrasing to the more concise form of "Figure 5 shows …"

L330: "that can be more appreciated" → "that is more pronounced"

L351: "appreciate" → "identify"

L368: "We can observe" → "We observe"

L379ff: Excessive use of the passive tense, consider rewording using an active form of "We investigated the influence …" or similar in at least part of the sentences.

---

## Author Response (AR2)

Dear Reviewer,

Thank you for the careful and comprehensive review of the manuscript. The main changes performed on the manuscript have been:

- The inversion of the color ramp in Figure 8 and 10.

- Summary rewording.

In addition to those changes, a number of corrections and clarifications have been made throughout the text regarding other minor concerns. Please, find below point-by-point responses (in blue) to your comments and suggestions.

Kind regards,

Javier Roger, on behalf of the authors

**Response to comments – Reviewer: Marvin Knapp**

**Major Comments:**

None

**Minor Comments:**

Line 139: "Different original values, characterized by diverse deviations, could have also led to overestimated dXCH4 values. Therefore, deviations from the model will probably introduce biases in the retrieved values." – As I understand it, I wouldn't call this a bias. The calculated enhancements just include the noise of the underlying data, as is to be expected. In the case of a two-channel retrieval, this noise is high, but it reduces with increasing information content from more channels (up to a point, as you illustrate in Figure 4). I suggest rephrasing to something like "Natural enhancements scatter around their true value since the measurements are subject to noise, thus an overestimation would have been equally likely."

Corrected.

L156: "… which leads to a background noise reduction but also an underestimation of enhanced pixels." – You've shown that with your plot, no need to use "could".

Corrected.

L159: I think you should state here clearly that artifact reduction is a major selling point of the retrieval. You should refer that it is a result of your investigation - something along the lines of "New retrieval artifacts may appear when including the 1700 nm absorption window, but overall, the increased spectral interval mitigates false detections efficiently, as is shown in section 3."

Corrected.

L165: Since the 0-radiance pixels between 1800-1950 nm do not cause an increase in variability, I am not convinced that your explanation is correct. Does the retrieval noise shrink again when you

exclude the water bands? In any case, it is reasonable to exclude these intervals, they may cause trouble, and do not add value to the retrieval. But it seems to me more like the non-zero values below 1400 nm cause the rise in uncertainty, which is somehow interesting. If you have strong evidence that it is the water bands (like the decrease of noise after exclusion), state it explicitly. If not, just note that you are removing the water bands for the abovementioned reasons and that the noise increase is due to the addition of the bright channels below 1400 nm.

Radiance pixel values around 1900 nm did not reach 0-values, but values around 1400 nm did. However, if I exclude those bands with 0-values from the ~1400 nm bands, the retrieval noise shrinks again. Here we show the left panel of Figure 4 when excluding those bands.

[Figure]

Therefore, the presence of 0-values is probably the reason of the sudden increase in retrieval noise. We clarify this in the text.

L176: Since the SWIR range from 1000-2500 nm shows a higher noise, why didn't you choose the interval from 1500-2500 nm for the COMBO-MF? I guess the even broader interval might be more capable of reducing artifacts, even though it has a larger variability. If you chose it because of that, you should state it here to avoid confusion.

Regarding the previous comment, if we remove the bands around 1400 nm, we find that there is no increase of variability. However, as the reviewer commented, the main reason has been the greater capability to exclude retrieval artifacts. We clarify this in the text.

Caption Figure 8: Add the subscript CH4 to the Q in the caption. Also, you should mention in the caption that the constant enhancement of CO2 and H2O is only added to the plume pixels.

Corrected.

Figure 8+10: You might consider flipping the colorbar of the red-to-blue differences, it is more common to denote positive values with red and negative with blue.

Corrected.

L353: "Therefore, an appropriate use of Combo-MF should take into account the surface composition beneath potential methane emissions." – This is hard to accomplish in reality, right? In any case, you can be more optimistic here. It is also possible that the COMBO-MF removes the influence of the air pollutants, which seem to significantly increase noise in the 2300-MF.

We write a sentence to be more optimistic regarding the application of Combo-MF. Regarding the influence of the air pollutants, we are not sure that the higher background noise in 2300-MF is due to the presence of air pollutants. Therefore, we do not mention it.

Figure 11: I think you should state here one time clearly that all shown plumes pass your plume masking procedure, and a-d are taken out as examples.

Corrected.

L380: I suggest adding a sentence which gives a broad overview, like "We investigated the retrieval performance in real observations of spectral imagers on satellites and airplanes, both using real and simulated plumes, and found a significant increase in the plume identification capability." The investigation of the retrieval using a broad spectrum of scenes and instruments was done thoroughly and detailed in this study, and should be highlighted in the beginning of the summary.

Added.

L390: You don't mention that you recommend he Mix-MF for emission quantification. You should mention it in the summary, and it fits when you point out the caveats of the COMBO-MF.

Mix-MF was applied using preliminary masks coming from the simulated plumes, so we have not assessed its performance in realistic conditions. Mix-MF was only used to evaluate the clutter removal (already stated in the summary: 'is less affected by background noise for small flux rate values') with the use of this preliminary mask. Therefore, we are not able to conclude if there is an improved performance in real cases. Then, we do not mention it in the text.

**Technical corrections:**

L156: make*s* the

Corrected.

L203: "Moreover, in Figure 5 we can see …" is wordy and somehow colloquial. I suggest rephrasing to the more concise form of "Figure 5 shows …"

Corrected.

L330: "that can be more appreciated" → "that is more pronounced"

Corrected.

L351: "appreciate" → "identify"

Corrected.

L368: "We can observe" → "We observe"

Corrected.

L379ff: Excessive use of the passive tense, consider rewording using an active form of "We investigated the influence …" or similar in at least part of the sentences.
Corrected.